# The stretch–shortening cycle effect is not associated with cortical or spinal excitability modulations

Lea-Fedia Rissmann[1], Brent James Raiteri[1,2], Wolfgang Seiberl[3], Tobias Siebert[4] and Daniel Hahn[1,2]

[1] *Human Movement Science, Faculty of Sport Science, Ruhr University Bochum, Bochum, North Rhine-Westphalia, Germany*
[2] *School of Human Movement and Nutrition Sciences, University of Queensland, Brisbane, Queensland, Australia*
[3] *Institute of Sport Science, Department of Human Sciences, Universität der Bundeswehr München, Neubiberg, Germany*
[4] *Department of Motion and Exercise Science, University of Stuttgart, Stuttgart, Germany*

Handling Editors: Richard Carson & Ross Pollock

The peer review history is available in the Supporting Information section of this article (https://doi.org/10.1113/JP287508#support-information-section).

*The Journal of Physiology*

**Abstract figure legend** Active muscle stretch before shortening increases force, work and power output compared with shortening only, which is known as the stretch–shortening cycle (SSC) effect. We explored whether neural modulations triggered by active muscle stretch persist during the SSC shortening phase, thereby potentially contributing to the SSC effect. Participants performed EMG amplitude matched submaximal voluntary shortening and SSC contractions with their plantar flexor muscles on a motor-driven dynamometer. During shortening, corticospinal and spinal excitability were assessed by transcranial magnetic stimulation and electrical stimulation of the cervicomedullary junction, respectively. Additionally, stretch reflex responses during active muscle stretch of the SSC contractions were assessed by EMG. We observed a significant SSC effect of 12%, but no stretch reflex responses during SSC stretch and no modulation of corticospinal or spinal excitability during SSC shortening in comparison with shortening only. This indicates that mechanical rather than neural mechanisms contribute to the SSC effect.

This article was first published as a preprint: Rissmann L-F, Raiteri BJ, Wolfgang Seiberl W, Siebert T, Hahn D. (2024). The stretch-shortening cycle effect is not associated with cortical or spinal excitability modulations. bioRxiv. https://doi.org/10.1101/2024.08.19.608542

The Journal of Physiology

**Abstract** It is unclear whether cortical and spinal excitability modulations contribute to enhanced stretch–shortening cycle (SSC) performance. Therefore, this study investigated cortical and spinal excitability modulations during and following shortening of SSC contractions compared with pure shortening (SHO) contractions. Participants ($n = 18$) performed submaximal voluntary plantar flexion contractions while prone on the dynamometer bench. The right foot was strapped onto the dynamometer's footplate attachment, and the resultant ankle joint torque and crank arm angle were recorded. Cortical and spinal excitability modulations of the soleus muscle were analysed by eliciting compound muscle actional potentials via electrical nerve stimulation, cervicomedullary motor-evoked potentials (CMEPs) via electrical stimulation of the spinal cord, and motor-evoked potentials (MEPs) via magnetic stimulation of the motor cortex. Mean torque following stretch was significantly increased by $7 \pm 3\%$ ($P = 0.029$) compared with the fixed-end reference (REF) contraction, and mean torque during shortening of SSC compared with SHO was significantly increased by $12 \pm 24\%$ ($P = 0.046$). Mean steady-state torque was significantly lower by $13 \pm 3\%$ ($P = 0.006$) and $9 \pm 12\%$ ($P = 0.011$) following SSC compared with REF and SHO, respectively. Mean steady-state torque was not significantly different following SHO compared with REF ($7 \pm 8\%$, $P = 0.456$). CMEPs and MEPs were also not significantly different during shortening of SSC compared with SHO ($P \geq 0.885$) or during the steady state of SSC, SHO and REF ($P \geq 0.727$). Therefore, our results indicate that SSC performance was not associated with cortical or spinal excitability modulations during or after shortening, but rather driven by mechanical mechanisms triggered during active stretch.

(Received 16 August 2024; accepted after revision 30 April 2025; first published online 16 June 2025)

**Corresponding author** D. Hahn: Human Movement Science, Faculty of Sport Science, Ruhr University Bochum, Bochum, North Rhine-Westphalia, Germany. Email: daniel.hahn@rub.de

**Key points**

- A stretch–shortening cycle (SSC) effect of 12% was observed during EMG-matched submaximal voluntary contractions of the human plantar flexors.
- The SSC effect was neither associated with cortical or spinal excitability modulations nor with stretch-reflex activity.
- The SSC effect was likely driven by mechanical mechanisms related to active muscle stretch, which have long-lasting effects during shortening.
- Residual force depression following SSC was not attenuated by the long-lasting mechanical mechanisms triggered during active muscle stretch.
- Steady-state torques were lower following shortening of SSCs *versus* pure shortening and fixed-end contractions at the same final ankle joint angle, but the torque differences were not correlated with cortical or spinal excitability modulations.

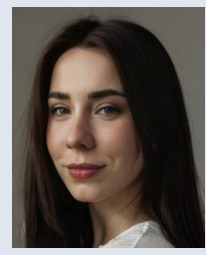

**Lea-Fedia Rissmann** earned her Bachelor's degree in Biology and Sports Science and her Master's degree in Sports and Exercise Sciences for Health and Performance at the Ruhr University Bochum. Her research focuses on neuromechanics and motor control, particularly how the cortex and spinal cord controls human movement. She studies the interaction between neural and mechanical factors in motor control, including the stretch–shortening cycle and its effects on performance and rehabilitation. In the future, she aims to explore sensory integration and signal processing in the human cortex to better understand how human movement is controlled.

## Introduction

During everyday locomotion, lower limb muscle–tendon units (MTU) often actively stretch before actively shortening, which is known as a stretch–shortening cycle (SSC). SSCs are unique because torque, work and power outputs during and after the shortening phase are typically higher than shortening contractions without a preceding stretch (SHO) (Cavagna et al., 1968; Seiberl et al., 2015), which is known as the SSC effect.

The SSC effect has been explained by mechanical and neural factors that are triggered by the stretch preceding the shortening (Seiberl et al., 2021). As the SSC effect has been observed at the single-fibre level, mechanical factors are at least partly located in the sarcomeres: during active muscle stretch, altered cross-bridge kinetics and increased passive forces contribute to the increased force capacity of muscles, called transient force enhancement (tFE; Bakenecker et al., 2020). tFE could subsequently cause higher forces (Edman et al., 1978; Katz, 1939; Leonard et al., 2010) and greater elastic energy return during successive shortening. Additionally, a phenomenon known as residual force enhancement (rFE; Abbott & Aubert, 1952; Edman et al., 1982) might also enable higher forces during and following shortening (Fukutani et al., 2015a; Hahn & Riedel, 2018; Seiberl et al., 2015). As rFE is thought to be caused by increased cross-bridge forces and the engagement of the elastic element titin (Herzog et al., 2016; Hessel et al., 2021; Rode et al., 2009; Rassier, 2012), these mechanisms within the sarcomere likely contribute to the SSC effect as well (Fukutani et al., 2017; Groeber et al., 2019; Hahn et al., 2023; Tomalka et al., 2020).

In addition to mechanical factors at the sarcomeric level neural mechanisms could also contribute to the SSC effect. For example, stretch reflexes that are triggered during active muscle stretch, by excitatory afferent muscle spindle discharges, could facilitate and synchronize the activation of the motor units during the shortening phase of the SSC (Trimble et al., 2000). This reflex activation could thus contribute to the SSC effect by increasing muscle activation compared with pure shortening contractions (Taube, Leukel, Gollhofer, 2012; Trimble et al., 2000; van Ingen Schenau et al., 1997). In contrast, the SSC effect might be attenuated by inhibitory mechanisms that are triggered during active muscle stretch, reducing forces during successive shortening, opposing the SSC effect (Cronin et al., 2011; Davey et al., 1994; Duchateau & Enoka, 2016; Westing et al., 1991). However as some studies did not find any neural inhibition during active muscle stretch (e.g. Hahn et al., 2012; for review, see Hahn, 2018), it is unclear whether there is a net inhibitory or net excitatory effect during active muscle stretch, which might originate from spinal and/or cortical sites.

To test excitability modulations at spinal and cortical sites, Gruber et al. (2009) elicited motor-evoked potentials (MEPs) and cervicomedullary motor-evoked potentials (CMEPs) in the human biceps brachii and brachioradialis muscles. These authors found smaller MEPs and CMEPs during active muscle stretch compared with fixed-end contractions, but increased MEP-to-CMEP ratios, which can be interpreted as increased cortical excitability during active muscle stretch. Further, it was shown that descending cortical drive contributes to early muscle activation during hopping SSCs, which was attributed to spinal stretch reflex activation (Zuur et al., 2009). Cortical and spinal excitability modulations might not just occur during active muscle stretch: excitability modulations might persist following active muscle stretch during shortening and the isometric hold phase (i.e. steady state) following the shortening phase of SSCs. In the steady state following active muscle stretch, several studies found contrasting results. Hahn et al. (2012) found increased cortical but unchanged spinal excitability and Sypkes et al. (2018a) found unchanged cortical but reduced spinal excitability. Furthermore, Frischholz et al. (2022) found that rFE was unaffected by corticospinal excitability modulations. However, this study did not assess whether corticospinal excitability was modulated during active muscle stretch or whether cortical and spinal excitability were modulated differently following stretch.

While active muscle stretch can trigger rFE, active muscle shortening can trigger residual force depression (rFD), which refers to the long-lasting reduction in muscle force following shortening (Abbott & Aubert, 1952). In the steady state following active muscle shortening, Grant et al. (2017) found unchanged MEPs and unchanged CMEPs in the presence of rFD during maximal voluntary dorsiflexion contractions, which can be interpreted as absent cortical and spinal excitability modulations. In contrast, Sypkes et al. (2018b) found decreased cortical, but increased spinal excitability in the presence of rFD during submaximal voluntary dorsiflexion contractions.

Based on the contradictory findings regarding potential neural modulations triggered by active muscle stretch and shortening, no clear statement can be made as to whether cortical and/or spinal excitability is altered by active muscle stretch and shortening. Further, to the best of our knowledge, no data exists on whether the combination of active muscle stretch and shortening modulates cortical and/or spinal excitability during and following SSC contractions. Therefore, the aim of this study was to investigate cortical and spinal excitability during and following the shortening phase of SSCs compared with pure shortening contractions during submaximal voluntary plantar flexion (PF) contractions. By doing this, we aimed to gain insight into possible neural mechanisms that contribute to the SSC effect.

We expected that (1) the stretch in our SSC condition would lead to tFE and rFE and facilitate the SSC effect, and therefore we recorded torque following stretch and during the steady state of a reference (REF$_{DF}$) and pure stretch (STR) contraction. We further expected (2) an increased torque during the shortening phase of SSC compared with SHO and reduced or similar rFD during the steady state following the shortening phase of SSC compared with SHO when the soleus EMG amplitudes were matched. Finally, we expected that (3) possible stretch-triggered changes in cortical and/or spinal excitability would be long-lasting and that these changes would be visible during the shortening and steady-state phases of SSC compared with SHO and reference (REF$_{PF}$) when EMG amplitudes were matched.

## Methods

### Participants

Eighteen recreationally active healthy adults (7 women, 11 men, 25.9 ± 5.6 years, 1.79 ± 0.05 m and 75.0 ± 10.5 kg) gave written informed consent prior to participating in this study. All subjects were free of neuromuscular diseases and musculoskeletal injuries for at least 6 months prior to and throughout the study. In addition, participants with a known increased risk of epileptic seizures, pregnant women, participants with implanted biomedical devices and/or electronic implants (e.g. brain pacemakers, heart pacemakers) or metal particles in the skull were excluded from participation. Ethics were approved by the Human Ethics Committee of Ruhr University Bochum (EKS V 16/2021) and all procedures were in accordance with the principles of the *Declaration of Helsinki*.

### Torque, angle and EMG

Resultant ankle joint torque and crank arm angle were recorded from the right foot of the participants' leg, while participants laid prone on the bench of a dynamometer (IsoMed2000, Ferstl GmbH, Germany) (Fig. 1*A*). The right foot was tightly strapped onto a footplate attachment to minimize heel lift during contractions (Fig. 1*B*). Hip and knee joint angles remained fully extended, and accessory movements during contractions were minimized by securing the participants' waist with a belt (Frischholz et al., 2022). Further, particular attention was paid to monitoring the participants' posture during the test and in avoiding head rotations; therefore, participants hyperextended their lower back and propped their head up on their hands. The lateral malleolus was aligned with the axis of rotation of the dynamometer in a neutral position (i.e. foot plate perpendicular to the shank), which we defined as 0° ankle joint angle. The range of motion was set from −10° PF to +15° dorsi flexion (DF) (Fig. 1*A*). Stretch velocity was set to 40° s$^{-1}$, and shortening velocity was set to 120° s$^{-1}$, while angular acceleration and

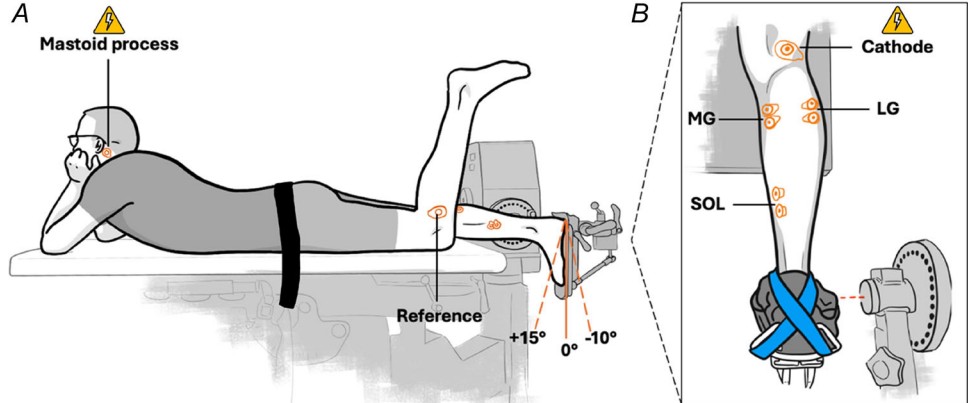

**Figure 1. Participant set-up on the dynamometer and electrode placement on the right triceps surae**
Participant laid prone on the bench of a dynamometer with the right foot strapped onto a footplate attachment. The participants' waist was fixed with a belt. Neutral position (i.e. foot plate perpendicular to the shank) was defined as 0°. The range of motion was set from −10° plantarflexed (PF) position to +15° dorsiflexed (DF) position. Knee and hips were fully extended. Participants had to keep their head up and supported in their hands during all contractions. The reference electrode for recording muscle activity and responses to stimulations was placed on the left fibular head. *A*, electrodes for spinal cord stimulation (voltage symbol) at the cervicomedullary junction were placed over the grooves behind the mastoid processes, with the cathode on the left side of the head. *B*, electrode placement for the recording of muscle activity and responses to stimulations of SOL, MG and LG. Electrode placement is also shown for the cathode for electric nerve stimulation (voltage symbol) within the popliteal fossa. The lateral malleolus was aligned with the axis of the dynamometer. A strap was used to firmly fix the foot to the footplate attachment. [Colour figure can be viewed at wileyonlinelibrary.com]

deceleration was set to 2000° s$^{-2}$. Resultant ankle joint torque and crank arm angles were sampled at 1000 Hz and were filtered using a dual low-pass fourth-order 20 Hz Butterworth filter and a dual low-pass fourth-order 6 Hz Butterworth filter, respectively.

Soleus (SOL), medial gastrocnemius (MG), lateral gastrocnemius (LG) and tibialis anterior (TA) muscle activities were recorded at 5000 Hz using surface electromyography (EMG) (Fig. 1*B*). EMG signals were pre-amplified 1000 times (NL844, Digitimer, Welwyn Garden City, UK) and high-pass filtered at 10 Hz (NL135 and NL144, Digitimer) using an analog filter. EMG signals were recorded using two electrodes (8 mm recording diameter, Ag/AgCl, Covidien, Mansfield, MA, USA) placed at an interelectrode centre-to-centre distance of 2 cm according to international guidelines with a single ground electrode of the same type attached to the fibular head of the left leg (Fig. 1*A*). The skin was prepped by shaving and abrading (Nuprep, Weaver and Company, Aurora, CO, USA) prior to wiping with alcohol-based disinfectant (Sterillium, BODE Chemie GmbH, Hamburg, Germany) to minimize conductive resistance in the skin (Hermens et al., 1999). All data were synchronized using a 16-bit A/D card within a Power1401 data acquisition interface with Spike2 software (CED, Cambridge, UK). For visualization, SOL EMG was smoothed with moving root mean square (RMS) over a 250 ms window and normalized to 40% maximal voluntary activity (MVA) of $EMG_{SOL}$ recorded during maximal voluntary contractions (MVC) at PF and DF.

## Stimulation techniques

Peripheral electrical nerve stimulation (ENS) was used to evoke M-waves. SOL M-waves were evoked by 1-ms single-pulse stimulations of the tibial nerve in the popliteal fossa (DS7AH, Digitimer) (Figs 1*B* and 2). Current passed from a cathode (Ag/AgCl electrode, Tyco Healthcare, Neustadt a.d. Donau, Germany) placed within the popliteal fossa over the tibial nerve to an anode (coal rubber pad, 10.264.6 cm, Empi, St. Paul, MN, USA) positioned 2 cm proximal to the patella. The stimulation site providing the greatest M-wave peak-to-peak amplitude was first located by a hand-held motor point pen (0.5 cm diameter; Compex, Hallbergmoos, Germany) in PF at rest. Once the optimal stimulation site was determined, the stimulation electrode was firmly fixed to this site with tape. M-wave shape and peak-to-peak amplitude were checked for clear visibility at 0° and PF during isometric contraction at 40% $EMG_{SOL}$. The intensity used to evoke a maximal M-wave (Mmax) of SOL during voluntary activation was multiplied by a factor of 1.2 for supramaximal motor nerve stimulation during all contractions.

Cervicomedullary electrical stimulation (CES) was used to evoke CMEPs in SOL (Figs 1*A* and 2). CMEPs were obtained after a 100-µs single-pulse stimulation of the spinal cord at the cervicomedullary junction using a constant voltage stimulator (D185, Digitimer). Electrodes (Ag/AgCl electrode, Tyco Healthcare) were placed over the grooves behind the mastoid processes, with the

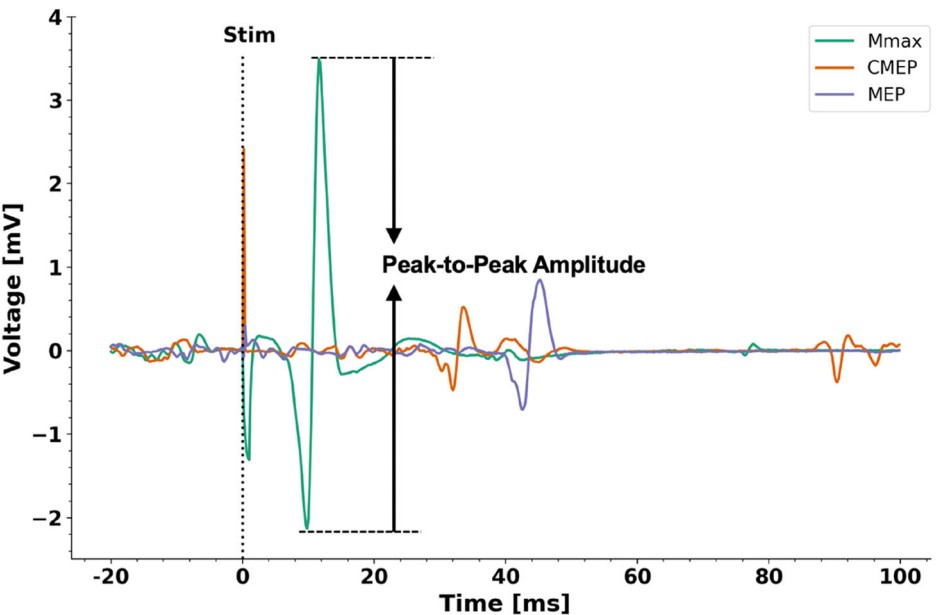

**Figure 2. Exemplary Mmax, CMEP and MEP voltage–time responses**
Individual raw EMG amplitude–time traces following stimulation from one representative participant in mV (individual muscle gains were accounted for). Mmax (green), CMEP (orange) and MEP (purple). The instant of stimulation is indicated by 'Stim'. The size of the responses was calculated as the peak-to-peak amplitude. [Colour figure can be viewed at wileyonlinelibrary.com]

cathode on the left side of the head. Stimulation intensities (350–800V) were matched to 10–30% of Mmax, while subjects maintained a submaximal contraction at 40% $EMG_{SOL}$. CMEPs had to be clearly visible compared with the background $EMG_{SOL}$. Due to the discomfort evoked by this kind of stimulation, stimulation intensity was not further increased from the amount each participant was able to withstand. However, if participants were unable to sustain the stimulation intensities required to elicit a response equivalent to at least 10% of Mmax, they were excluded from the study.

Transcranial magnetic stimulation (TMS) was used to evoke MEPs in SOL (Fig. 2). MEPs were elicited by a 280-µs single-pulse magnetic stimulation of the motor cortex by a double-cone coil (D-B80, MagPro Compact, MagVenture, Farum, Denmark). The coil was placed in parallel and in an anterior–posterior direction to the tangent plane of the skull, thereby inducing an eddy current perpendicular to the coil and in a posterior–anterior direction in the cortical motor area of the soleus. The cortical motor area of the soleus was assumed to be slightly left side of the vertex, which was determined through measurement of the head and then defined as the initial starting position for coil placement. Once the initial starting position was defined, several stimuli were delivered at different locations surrounding the initial starting position until the hotspot was found. The hotspot was defined as the spot where the largest MEP peak-to-peak amplitude in SOL was evoked by a single stimulation while subjects performed submaximal steady-state isometric contractions in PF at 40% $EMG_{SOL}$. The hotspot was then marked on the skin with a stencil marker (Purple Stencil Marker[TM] Surgical Skin Scribe, Stencil Stuff, USA) for replication throughout the experiment. Replication was warranted by drawing the outer shape of the coil on the scalp. This created a V shape measuring approximately 5 cm, which made it possible to position the coil as accurately as possible on the head. Stimulation intensity was adjusted until the peak-to-peak amplitudes of the MEPs were closely matched to those of the CMEPs.

## Protocol

Prior to testing, participants had to perform at least three familiarization sessions. During the familiarization sessions, participants had to practice all contraction conditions and were familiarized with ENS and CES. As CMEPs in lower limb muscles could not be observed in every subject (Taylor & Gandevia, 2004), non-responders (20 out of 38) were excluded during the familiarization sessions.

Participants that were included were asked to avoid any training of the lower limbs 2 days prior to the actual test session, as well as caffeine intake on the same day. At the beginning of the test session, participants had to perform a warm-up including ten, five, three and two contractions at an intensity of 30%, 50%, 70% and 90% of perceived maximum effort, respectively. Afterwards, maximal voluntary PF torque and MVA of SOL were assessed by 2–3 MVCs in PF and DF each. Only trials with a plateau in torque and $EMG_{SOL}$ were used for further testing, which was verified by visual inspection. Trials with the highest MVA were used to match %$EMG_{SOL}$-based contractions.

To adjust the necessary stimulation intensities for testing, participants received visual feedback of the $EMG_{SOL}$ on a computer monitor and were required to perform contractions at 40% $EMG_{SOL}$ in PF in a specified order: First, Mmax was determined as a reference to adjust the sizes of the CMEPs and the MEPs to a corresponding %Mmax. Second, the stimulation intensity of CES was set to obtain CMEPs, and third, based on the obtained CMEP size, the stimulation intensity of TMS was set to obtain MEPs matched to CMEP size.

Following the set-up of the stimulation intensities, the testing started, which involved five different contraction conditions. All five contraction conditions were $EMG_{SOL}$ controlled, which was provided as visual feedback and included the following: two fixed-end voluntary reference contractions at PF ($REF_{PF}$) and DF ($REF_{DF}$) and three dynamic voluntary contractions involving pure shortening (SHO), stretch–shortening (SSC) and pure stretch (STR), which were time-matched (Fig. 3*A* and *B*). All contraction conditions (with exception of SHO) were performed with 40% $EMG_{SOL}$ of the corresponding ankle joint angle. SSC and STR contractions started with a 2-s preactivation of 40% $EMG_{SOL}$ before the onset of joint rotation. To match $EMG_{SOL}$ during the shortening phase of SHO and SSC contractions, SHO contractions started with a slightly higher ankle joint-angle specific pre-activation of 47% $EMG_{SOL}$, which lasted approximately 2.5 s before joint rotation started so that shortening of SHO and SSC was time-matched (see Fig. 3*E*). All dynamic voluntary contractions conditions (SHO, SSC, STR) were followed by an isometric hold phase at $REF_{DF}$ or $REF_{PF}$, respectively (Fig. 3*C* and *D*).

Further, contraction conditions were then divided into 'stimulated' (i.e. $REF_{PF}$, SHO and SSC) and 'non-stimulated' (i.e. $REF_{DF}$ and STR) conditions. During the stimulated conditions, subjects received ENS, CES and TMS superimposed on the voluntary contractions (Fig. 3*C*). During $REF_{PF}$, subjects received only one stimulation, while during SHO and SSC, subjects received two stimulations. Stimulation timing was set during the shortening phase of SHO and SSC, at the time when the ankle joint passed 0° ankle angle and 3 s after the end of the shortening phase of SHO and SSC, and at time-matched instants during $REF_{PF}$. During

STR and REF$_{DF}$ participants received no stimulations (Fig. 3*D*).

All contraction conditions were randomized in blocks based on the type of stimulation (i.e. ENS, CES and TMS) and were further divided into 'stimulated' and 'non-stimulated' conditions. All 'stimulated' and all 'non-stimulated' conditions were then further randomized based on the contraction condition. Participants had to perform three repetitions of each contraction condition for ENS, five repetitions for CES and 10 repetitions for TMS for the 'stimulated' conditions. For the 'non-stimulated' conditions, subjects only had to perform three contractions per contraction condition. This resulted in a total of 54 submaximal contractions with superimposed stimulation and six submaximal contractions without superimposed stimulation. Participants thus received a total of 15 nerve stimulations, 25 spinal cord stimulations and 50 motor cortex stimulations. To prevent fatigue, participants had to take at least a 2-min break between contractions and longer if requested. Participants rested in PF as it was a more comfortable position.

## Data analysis

Mean resultant torque during the shortening phase of SHO and SSC was calculated over a 105 ms time window, when shortening speed and angular displacement was constant (Fig. 3*C*). The SSC effect was then calculated as the percentage difference in mean torque between SHO and SSC during this time window.

During the steady state of REF$_{PF}$, SHO and SSC, as well as REF$_{DF}$ and STR, the mean resultant torque was calculated over a 500 ms time window 1 ms prior to the second stimulation instant (Fig. 3*C* and *D*). tFE was calculated as the percentage difference in peak torque during the SSC stretch phase compared with the time-matched torque prior to shortening of SHO. rFD or rFE were calculated as the percentage difference in mean resultant torque between REF$_{PF}$ and SHO (rFD), between REF$_{PF}$ and SSC (rFD) and between REF$_{DF}$ and STR (rFE).

To determine Mmax, CMEP and MEP peak-to-peak amplitudes within each participant, a cross-correlation was performed between the averaged trials and the individual trials of each contraction condition for each muscle separately. Cross-correlations were performed over physiologically relevant time windows of 5 to 25 ms

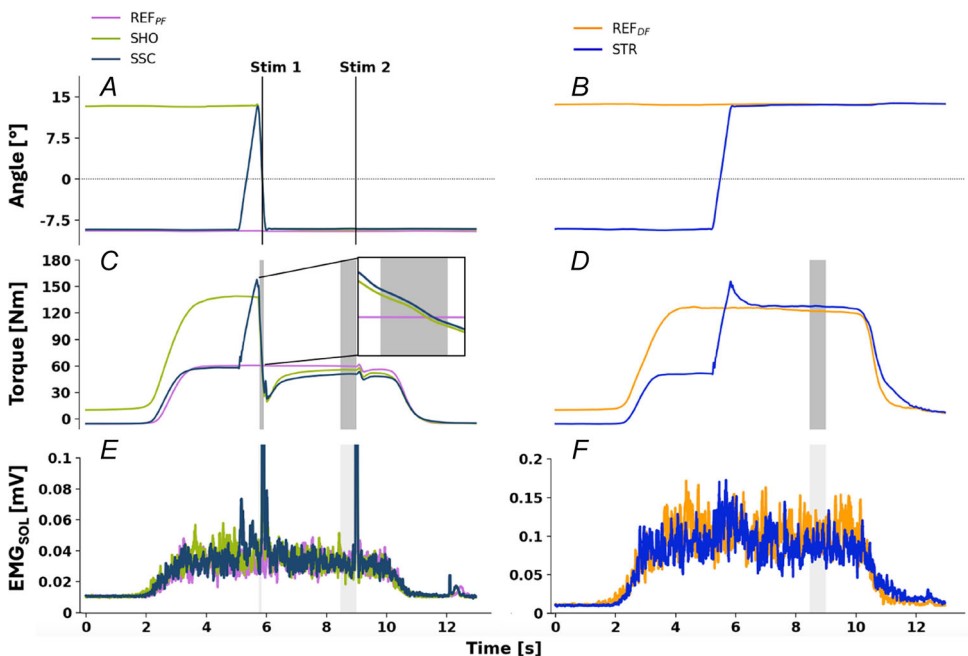

**Figure 3. Mean crank arm angle–time (*A* and *B*), resultant torque–time (*C* and *D*) and root mean squared (200 ms) soleus muscle activity (EMG$_{SOL}$) averaged from all participants during reference in plantarflexed position (REF$_{PF}$), pure shortening (SHO) and stretch–shortening (SSC) (*n* = 15) (*A*, *C* and *E*) and reference in dorsal flexed position (REF$_{DF}$) and pure stretch (STR) (*n* = 12) (*B*, *D* and *F*)**
*A*, the instant of stimulation during the shortening phase in SHO (green) and SSC (dark blue) is labelled Stim 1. The instant of stimulation during the steady state of REF$_{PF}$ (violet), SHO and SSC and is labelled Stim 2. *B*, no stimulation was applied during REF$_{DF}$ (orange) and STR (blue). *C* and *D*, the dark grey bars indicate the time period over which torque was averaged during shortening and during the steady state for each contraction condition. *E* and *F*, the light grey bars indicate the period over which EMG$_{SOL}$ was averaged during shortening and during the steady state for each contraction condition and each muscle. [Colour figure can be viewed at wileyonlinelibrary.com]

for Mmax, 20 to 50 ms for CMEPs and 30 to 60 ms for MEPs (Fig. 2). Trials with a correlation coefficient >0.8 were considered as valid and included in the analysis to determine the peak-to-peak amplitude (i.e. size) of each response (Fig. 2). The mean size of each response for each contraction condition was then calculated if there were at least two (Mmax), three (CMEP) and five (MEP) valid responses. This resulted in the inclusion of always 3 Mmax, 3–5 CMEP and 5–10 MEP responses per contraction condition per participant.

The mean Mmax size was used to normalize the sizes of the CMEPs and MEPs during the corresponding contraction condition. The normalized CMEPs and MEPs were then averaged within each contraction condition within each participant and compared between SHO and SSC during the shortening phase, and between $REF_{PF}$, SHO and SSC during the steady state following shortening.

EMG signals recorded from SOL, MG, LG and TA were smoothed using a moving RMS amplitude calculation. The moving RMS window had a step size of 1 data point (i.e. 0.2 ms), a duration of 50 ms and was applied from 99 to 1 ms before the first stimulation during the shortening phase of SHO and SSC (Fig. 3*E*). A different moving RMS window with a step size of 1 data point and a duration of 200 ms was applied from 699 to 1 ms before the second stimulation during the steady state of $REF_{PF}$, SHO and SSC (Fig. 3*E*). This same moving RMS window was applied over a time-matched period during $REF_{DF}$ and STR (i.e. from 2300 to 2999 ms after the crank arm rotation; Fig. 3*F*).

The smoothed $EMG_{SOL}$, $EMG_{MG}$, $EMG_{LG}$ and $EMG_{TA}$ of the contraction conditions including a stimulation ($REF_{PF}$, SHO and SSC) were normalized to Mmax of the corresponding contraction condition. The smoothed $EMG_{SOL}$, $EMG_{MG}$, $EMG_{LG}$ and $EMG_{TA}$ of the contraction conditions not including a stimulation ($REF_{DF}$, STR) were normalized to the MVA of the respective muscle that was measured during the plantar flexion MVC in DF.

To detect potential stretch reflex activity, the raw $EMG_{SOL}$, $EMG_{MG}$ and $EMG_{LG}$ data were rectified after direct current offset removal and then averaged over all trials from either the SSC or STR conditions within each participant. In both conditions, the start of the stretch phase was defined as the onset of the crank arm rotation (i.e. ankle dorsiflexion from PF to DF). The onset of crank arm rotation was defined as the last zero crossing of the crank arm position's first derivate before crank arm velocity became and remained positive. Stretch reflexes were then visually identified for each muscle separately at different latencies from the crank arm onset. The onset of the short-latency response (SLR) was defined as the first major deflection in the mean rectified EMG of each muscle (latency ∼45ms). The end of the SLR was identified as the second mean rectified EMG amplitude

peak (Ogiso et al., 2002). The subsequent EMG peaks were identified depending on their latency as mid-latency response (MLR, ∼70 ms), late latency response (LLR, ∼90 ms) and late latency response 2 ($LLR_2$, ∼120 ms) (Taube et al., 2008).

## Statistical analysis

The statistical analysis was performed with GraphPad Prism (version 10.0.3, GraphPad Software, Boston, MA, USA), and the $\alpha$-level was set to 0.05. Results are provided as means ± standard deviation. We used a two-way repeated-measures restricted maximum likelihood mixed-effects model to assess mean differences in normalized EMG amplitude (151 of 420 values are missing due to excluded trials) among contraction conditions and between muscles (contraction condition × muscle). Similarly, we used this model to assess mean differences in resultant torque (five missing values) among the contraction conditions and stimulation techniques (contraction condition × stimulation technique). We also used this model to assess mean differences in response size (i.e. normalized MEPs and CMEPs) among contraction conditions and stimulation techniques (contraction condition × stimulation technique; 26 missing values). We used Student's paired *t* test or a one-way repeated-measures ANOVA to assess mean differences in mean torque averaged across stimulation techniques (complete data missing of 2 and 3 participants) and Mmax (no missing values) among contraction conditions. Following a significant interaction (or a significant main effect for contraction condition when there were >2 levels), we conducted Holm-Šídák multiple comparisons to identify which conditions were significantly different. Further, a paired *t* test was used to determine whether there was a significant mean difference in tFE between SHO and SSC and a Wilcoxon test was used to determine the significant mean difference in PF torque during the steady state between $REF_{DF}$ and STR, since normal distribution was violated as indicated by a Shapiro–Wilk test.

Pearson's correlation coefficient was calculated to assess the strength of the relations between the SSC effect and (1) the delta cortical excitability or (2) the delta spinal excitability. Spinal excitability was defined as the normalized CMEP size, and cortical excitability was calculated as the difference between the normalized MEP and normalized CMEP sizes. Delta excitability was calculated as the difference in cortical or spinal excitability between SSC and SHO during shortening. Pearson correlation coefficients were also calculated to assess the strength of the relations between the SSC effect and (3) rFE and (4) the difference in rFD between SHO and SSC during the steady state. Finally, Pearson's

**Table 1. Normalized EMG amplitudes of the soleus (SOL), medial gastrocnemius (MG), lateral gastrocnemius (LG) and tibialis anterior (TA) muscles during the different contraction conditions**

| | During shortening | | During steady state | | | | |
| --- | --- | --- | --- | --- | --- | --- | --- |
| | SHO | SSC | $REF_{PF}$ | SHO | SSC | $REF_{DF}$ | STR |
| $EMG_{SOL}$ (%) | $1.5 \pm 0.6$ | $1.4 \pm 0.2$ | $1.1 \pm 0.4$ | $1.1 \pm 0.5$ | $1.1 \pm 0.5$ | $32.0 \pm 1.7$ | $33.3 \pm 2.3$ |
| $EMG_{MG}$ (%) | $2.0 \pm 0.6$ | $2.1 \pm 0.7$ | $1.7 \pm 0.7$ | $1.6 \pm 0.6$ | $1.6 \pm 0.6$ | $37.8 \pm 5.7$ | $37.6 \pm 5.9$ |
| $EMG_{LG}$ (%) | $2.0 \pm 0.8$ | $2.5 \pm 0.8$ | $2.0 \pm 1.0$ | $1.9 \pm 0.7$ | $2.1 \pm 1.0$ | $36.6 \pm 5.2$ | $36.4 \pm 5.2$ |
| $EMG_{TA}$ (%) | $0.6 \pm 0.2$ | $0.7 \pm 0.1$ | $0.6 \pm 0.1$ | $0.7 \pm 0.1$ | $0.7 \pm 0.1$ | $38.7 \pm 5.9$ | $37.4 \pm 4.5$ |

A two-way repeated-measures restricted maximum likelihood mixed-effects model was used to assess mean differences in normalized EMG amplitude. The mean normalized $EMG_{SOL}$, $EMG_{MG}$, $EMG_{LG}$ and $EMG_{TA}$ were not significantly affected by the contraction condition. Mean $\pm$ SD percentages are presented ($n = 15$). The mean EMG amplitude from each condition with stimulation was normalized to the maximum M-wave (Mmax) size obtained from the corresponding contraction condition from the corresponding muscle. The mean EMG amplitude from each condition without stimulation was normalized to the MVA values in the DF position of the corresponding muscle. Mean normalized EMG amplitudes were calculated during the shortening phase of SHO and SSC, as well as during the isometric steady state of $REF_{PF}$, SHO and SSC and $REF_{DF}$ and STR.

**Table 2. Resultant plantar flexion torques**

| | During shortening | | During steady state | | | | |
| --- | --- | --- | --- | --- | --- | --- | --- |
| | SHO | SSC | $REF_{PF}$ | SHO | SSC | $REF_{DF}$ | STR |
| Torque (N m) | $70.6 \pm 24.7$ | $79.1 \pm 18.6^*$ | $55.9 \pm 17.0$ | $53.7 \pm 17.8^*$ | $48.6 \pm 16.4^{\#}$ | $117.6 \pm 29.3$ | $126.2 \pm 28.0^*$ |

Student's paired *t* test (during shortening: SHO *vs.* SSC), a one-way repeated-measures ANOVA (during steady state: $REF_{PF}$ *vs.* SHO *vs.* SSC) and a Wilcoxon test (during steady state: $REF_{DF}$ *vs.* STR) were used to assess mean differences in resultant plantar flexion torque. Mean $\pm$ SD values are presented in N m ($n = 12$). Resultant mean torque was calculated during the shortening phase of SHO and SSC, as well as during the isometric steady state of $REF_{PF}$, SHO and SSC and $REF_{DF}$ and STR. Significant differences are indicated by an asterisk (*) or a hash (#).
*Significantly different compared with other condition.
#Significantly different compared with $REF_{PF}$ and SHO.

correlation coefficient was calculated to assess the strength of the relations between the difference in rFD ((5) $REF_{PF}$ *vs.* SHO, (6) $REF_{PF}$ *vs.* SSC) and delta cortical excitability or delta spinal excitability. No statistical test was performed on the data of the stretch reflex activity, as this was detected in one of the 15 participants only.

## Results

### EMG

The mean normalized $EMG_{SOL}$, $EMG_{MG}$, $EMG_{LG}$ and $EMG_{TA}$ were not significantly affected by the contraction condition but were significantly different among muscles. However, there was no significant interaction between contraction condition and muscle. Similar results were found during the shortening phase between SSC and SHO, as well as during the steady-state phase among SSC, SHO and $REF_{PF}$, and STR and $REF_{DF}$. These findings indicate successful matching of individual muscle activity levels throughout the experiment (see Table 1 for details).

### Torque

Torque was not significantly affected by stimulation technique (shortening, $F_{1.22,13.48} = 0.76$, $P = 0.422$; steady state, $F_{1.52,21.35} = 0.40$, $P = 0.618$), with no significant interaction between stimulation technique and contraction condition (shortening, $F_{1.35,11.55} = 0.31$, $P = 0.652$; steady state $F_{2.71,37.92} = 1.20$, $P = 0.313$). The mean torque during the shortening phase averaged across the stimulation techniques was significantly enhanced during SSC compared with SHO ($12.0\% \pm 24.5\%$, $P = 0.046$; Table 2 and Fig. 4*A*). The mean steady-state torque averaged across stimulation techniques was also significantly affected by contraction condition ($F_{1.22,17.13,} = 9.23$, $P = 0.005$; Table 2 and Fig. 4*B*). The mean steady-state torque was significantly lower following SSC compared with $REF_{PF}$ ($-13.1 \pm 3.1\%$, $P = 0.006$) and significantly lower compared with SHO ($-7.8 \pm 11.5\%$, $P = 0.011$). The mean steady-state torque was not significantly different following SHO compared with $REF_{PF}$ ($-7.2 \pm 8.2\%$, $P = 0.456$). tFE was significantly enhanced during SSC compared with SHO ($15.1\% \pm$

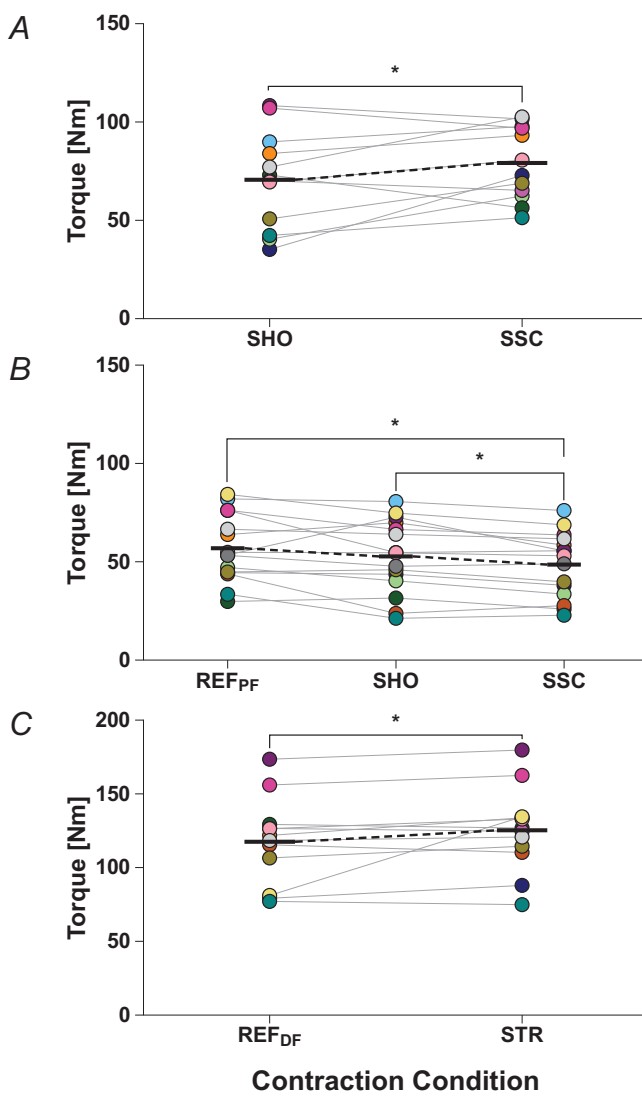

**Figure 4. Recorded plantar flexion torques among the contraction conditions**
Student's paired *t* test was used to assess mean differences in averaged torque between SHO and SSC during shortening (*A*), a one-way repeated-measures ANOVA to assess mean differences in averaged torque between REF$_{PF}$, SHO and SSC following shortening (*B*) and a Wilcoxon test to assess mean differences in torque between REF$_{DF}$ and STR (*C*). Torque during the shortening phase was significantly enhanced during SSC compared with SHO (*A*), significantly lower following SSC compared with REF$_{PF}$ (*B*) and compared with SHO (*C*) and significantly enhanced following STR compared with REF$_{DF}$. Individual data points are presented as filled circles, mean data points as black horizontal lines connected by broken lines, and significant differences between conditions (*P* ≤ 0.05) are indicated by an asterisk. Paired values are indicated by connecting lines and identical symbol colours. During shortening, *n* = 12 (*A*). During the steady state, *n* = 15 for REF$_{PF}$, SHO and SSC (*B*), and *n* = 11 for REF$_{DF}$ and STR (*C*). [Colour figure can be viewed at wileyonlinelibrary.com]

25.2%, *P* = 0.003). Additionally, the mean steady-state torque was significantly enhanced during STR compared with REF$_{DF}$ (7.3 ± 3.6%, *P* = 0.004; Table 2 and Fig. 4*C*).

## Responses

Mean SOL Mmax was not significantly different during the shortening phase between SSC and SHO (*P* = 0.454; Table 3 and Fig. 5*A*) or during the steady state among SSC, SHO and REF$_{PF}$ (*F* = 0.806, *P* = 0.442; Table 3 and Fig. 5*B*). Normalized SOL CMEP and MEP sizes were also not significantly different during the shortening phase between SSC and SHO (*F*$_{1,6}$ = 0.02, *P* = 0.885; Table 3 and Fig. 5*C* and *E*) or during the steady state among SSC, SHO and REF$_{PF}$ (*F*$_{2,10}$ = 0.328, *P* = 0.727; Table 3 and Fig. 5*D* and *F*). Stretch reflexes (SLR) were only visually identified in 1 of 15 participants from the SOL, but not from the MG or LG between 45 and 60 ms after the onset of crank arm rotation.

## Correlations

During the shortening phase, delta cortical excitability (CE%) and delta spinal excitability (SE%) did not significantly correlate with the SSC effect (CE%: *r* = 0.15 [95% CI: −0.52 to 0.71], *P* = 0.679; SE%: *r* = 0.16 [95% CI: −0.48 to 0.69], *P* = 0.628; Fig. 6*A* and *B*). There was also no significant correlation between the SSC effect during shortening and rFE following stretch (*r* = 0.611 [95% CI: −0.02 to 0.89], *P* = 0.060) (Fig. 6*E*) and rFD following shortening (*r* = −0.068 [95% CI: −0.61 to 0.52], *P* = 0.833) (Fig. 6*F*). Similarly, delta cortical excitability and delta spinal excitability did not significantly correlate with rFD for SHO (CE%: *r* = −0.52 [95% CI: −0.85 to 0.11], *P* = 0.097; SE%: *r* = 0.51 [95% CI: −0.09 to 0.83], *P* = 0.090; Fig. 6*C* and *D*) and rFD for SSC following shortening (CE%: *r* = 0.09 [95% CI: −0.57 to 0.68], *P* = 0.800; SE%: *r* = 0.23 [95% CI: −0.38 to 0.71], *P* = 0.456; Fig. 6*C* and *D*).

## Discussion

Our results indicate that the SSC effect is unlikely to be associated with cortical or spinal excitability modulations. This study was the first to investigate cortical and spinal excitability during and following the shortening phase of SSCs compared with pure shortening contractions. We found a significant SSC effect of ∼12.0%, but unchanged cortical and spinal excitability during the shortening phase of SSC contractions compared with pure SHO contractions. During the steady state following shortening, we found increased rFD following SSC compared with SHO and unchanged cortical and spinal excitability.

**Table 3. Response sizes following stimulation for the SOL muscle**

| Response | During shortening | | During steady state | | |
|---|---|---|---|---|---|
| | SHO | SSC | REF$_{PF}$ | SHO | SSC |
| Mmax (mV) | 2.7 ± 1.4 | 2.6 ± 1.2 | 1.9 ± 1.0 | 1.9 ± 0.9 | 2.0 ± 1.0 |
| CMEP/Mmax (%) | 8.4 ± 7.3 | 8.1 ± 4.9 | 13.2 ± 6.9 | 14.6 ± 6.7 | 14.0 ± 7.2 |
| MEP/Mmax (%) | 11.4 ± 5.2 | 11.2 ± 4.1 | 15.8 ± 9.0 | 19.6 ± 11.8 | 19.9 ± 11.1 |

Student's paired *t* test (during shortening: SHO *vs.* SSC) and a one-way repeated-measures ANOVA (during steady state: REF$_{PF}$ *vs.* SHO *vs.* SSC) were used to assess mean differences in Mmax among contraction conditions. Following a significant interaction we conducted a Holm–Šídák multiple comparisons test. Further a two-way repeated-measures restricted maximum likelihood mixed-effects model was used to assess mean differences in normalized CMEP and MEP responses. There were no significant differences found. The data are mean ± SD values of the maximum M-wave (Mmax) size (*n* = 15), the cervicomedullary motor-evoked potential (CMEP) size (*n* = 13 during shortening, *n* = 12 during steady state), and the motor-evoked potential (MEP) size (*n* = 14 during shortening, *n* = 13 during steady state). CMEP and MEP sizes were both normalized to Mmax of the respective contraction condition. Responses were obtained during the shortening phase of SHO and SSC, and during a subsequent steady-state phase of the same conditions. Responses were also obtained during a time-matched steady-state phase during REF$_{PF}$ at the same final ankle joint angle.

## Force enhancement during and following active muscle stretch

During and following active muscle stretch, muscle force capacity is increased compared with a fixed-end reference contraction at the same muscle length and muscle activity level, which was termed tFE and rFE, respectively (Bakenecker et al., 2020). We used a control condition (REF$_{DF}$ *vs.* STR) to assess whether our stretch conditions elicited tFE and rFE. As expected, we found a significant tFE of 15 ± 25% at the end of the stretch and a significant rFE of 7 ± 3% following stretch relative to REF$_{DF}$. This latter finding aligns with previous studies showing rFE *in vivo* in various muscles following various stretch conditions (Fukutani et al., 2017; Hahn et al., 2010; Herzog et al., 2006; Lee & Herzog, 2002; Pinniger & Cresswell, 2007; Rassier & Herzog, 2005; Seiberl et al., 2015). To ensure that our range of motion and selected pre-activation were sufficient to induce fascicle stretch in the soleus muscle, we collected ultrasound pilot data on *n* = 1 and observed soleus fascicle stretch of ∼10 mm. Therefore, it is reasonable to assume that the stretch amplitude and stretch velocity we selected were sufficient to trigger the mechanical mechanisms that result in tFE and rFE during and following active muscle stretch.

## The stretch–shortening cycle effect

As expected, we found that the mean torque during the shortening phase of SSC was significantly higher compared with SHO, and this resulted in a mean SSC effect of ∼12%, which supports our hypothesis that SSCs lead to enhanced performance during shortening. The existence of a SSC effect aligns with previous literature (Cavagna et al., 1968; van Ingen Schenau et al., 1997); however, the size of the SSC effect in our study (12 ±

24%) was smaller. This could be because previous studies used muscle belly or peripheral nerve stimulation rather than voluntary contractions and we used a higher pre-load for our pure SHO *versus* SSC contractions. Regarding potential mechanisms behind the SSC effect, a recent study (Goecking et al., 2024) showed that the increased torque at the onset of shortening during SSCs due to active muscle stretch (i.e. tFE and its underpinning mechanisms) contributed to the observed SSC effect. This also supports the early assumption by Cavagna et al. (1968) that the SSC effect is partly due to contractile components, which recently was further supported by work that found a SSC effect of up to 30% using skinned fibre preparations (Fukutani & Herzog, 2019; Fukutani et al., 2017; Tomalka et al., 2020, 2021).

## The history-dependence of muscle action

Based on the finding that rFE was not completely abolished following shortening during SSCs, it was hypothesized that tFE and rFE share some common mechanisms and that these stretch-induced mechanisms contribute to the SSC effect (Seiberl et al., 2015). This is why we obtained steady-state torques following SHO and SSC. Although we observed rFE following active muscle stretch, we found a significant rFD of ∼13% following SSC (*P* = 0.006), while the mean torque following SHO was similar compared with REF$_{PF}$ and significantly larger by ∼8% than the mean torque following SSC (*P* = 0.011) (Figs 3*C* and 4*B*). This is contrary to our expectations and to previous literature reporting rFE, non-significant rFD or reduced rFD following SSCs (Fortuna et al., 2017, 2018; Hahn and Riedel, 2018). Further, our results showed no significant correlation between the SSC effect and the magnitudes of rFE and rFD (Fig. 6*E* and *F*). These results suggest that under the given contraction conditions, the

stretch-induced mechanisms underpinning rFE did not contribute much or at all to the observed SSC effect. Further, larger rFD following SSC but similar torque following SHO compared with REF_PF suggests that the mechanisms underpinning rFD rather attenuated the SSC effect. As this is in contrast to similar previous studies using electrical stimulation, (Fukutani et al., 2015b, 2017; Hahn and Riedel, 2018), the type of muscle activation (i.e. electrically *vs.* voluntarily) might help to explain the contradictory results (Paternoster et al., 2021). Further, it was shown that the mechanisms underpinning rFE and their contribution to the SSC effect depend on the specific contraction conditions (i.e. shortening amplitude

and shortening speed) (Bakenecker et al., 2022; Fukutani et al., 2019; Holzer et al., 2024), which were not varied in this study.

## Stretch reflex during active muscle stretch

Apart from the mechanical contributions discussed above, stretch-reflex activity is a potential contributor to the SSC effect (Komi & Gollhofer, 1997; Turner & Jeffreys, 2010). However, in our study, SLR stretch reflexes were found in only 1 out of 15 participants for SOL, but not for MG or LG. This is in contrast to studies reporting a significant contribution to the SSC effect by stretch-reflex

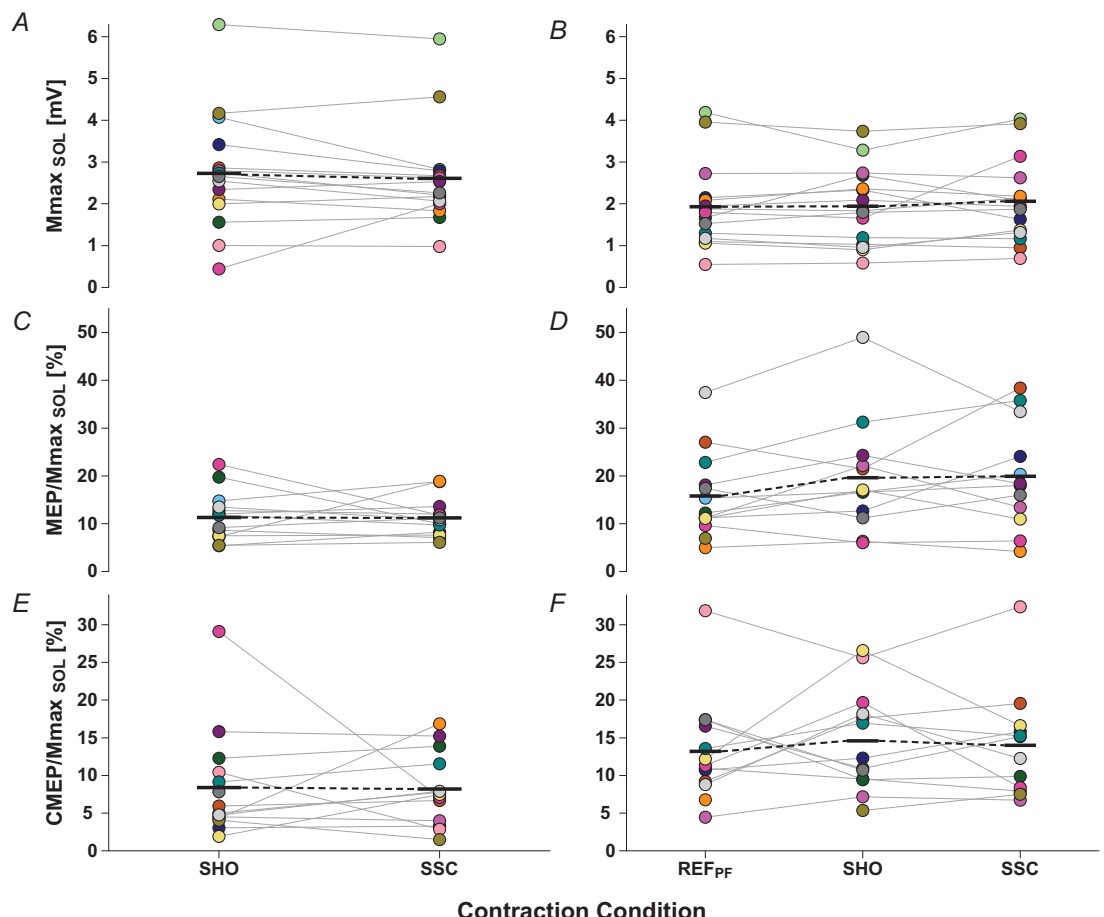

**Figure 5. Recorded responses obtained from soleus muscle during the contraction conditions including a stimulation**

Neither a paired *t* test (during shortening: SHO *vs.* SSC) nor a one-way repeated-measures ANOVA (during steady state: REF_PF *vs.* SHO *vs.* SSC) indicated significant differences in mean Mmax among contraction conditions (*A* and *B*). Further, a two-way repeated-measures restricted maximum likelihood mixed-effects model did not indicate mean differences in normalized MEP (*C* and *D*) or CMEP responses (*E* and *F*). Individual data points are presented as filled circles and mean data points as black horizontal lines connected by broken lines. Paired values are indicated by connecting lines and identical symbol colours. Mmax sizes are presented in mV as individual data points (*n* = 15) during shortening (*A*) and (*n* = 15) during the steady state (*B*). MEP and CMEP sizes were normalized to the Mmax size from the corresponding contraction condition. MEP/Mmax sizes are presented as a percetage as individual data points (*n* = 14) during shortening (*C*) and (*n* = 13) during the steady state (*D*). Similarly, CMEP/Mmax sizes are presented as a percentage as individual data points (*n* = 13) during shortening (*E*) and (*n* = 12) during the steady state (*F*). [Colour figure can be viewed at wileyonlinelibrary.com]

activity in human triceps surae muscles (Dietz, 1981, Zuur et al., 2009). A reason for the contrary findings might be that stretch-reflex activity was usually observed during SSC-type activities such as running, hopping, or landing (Dietz, 1981; Komi & Gollhofer, 1997; Schuster et al., 2020; Taube, Leukel, Lauber, et al., 2012), which involve much faster joint rotations compared with the controlled dynamometer conditions in this study. Accordingly, the stretch-reflex activity could possibly contribute to the SSC

effect in everyday movements, but this was not the case for our dynamometer-specific contraction conditions.

## Cortical and spinal excitability during muscle shortening

As rFE has been linked with modulations in cortical and spinal excitability (Gruber et al., 2009; Hahn et al., 2012; Sypkes et al., 2018a), it is also possible that neural

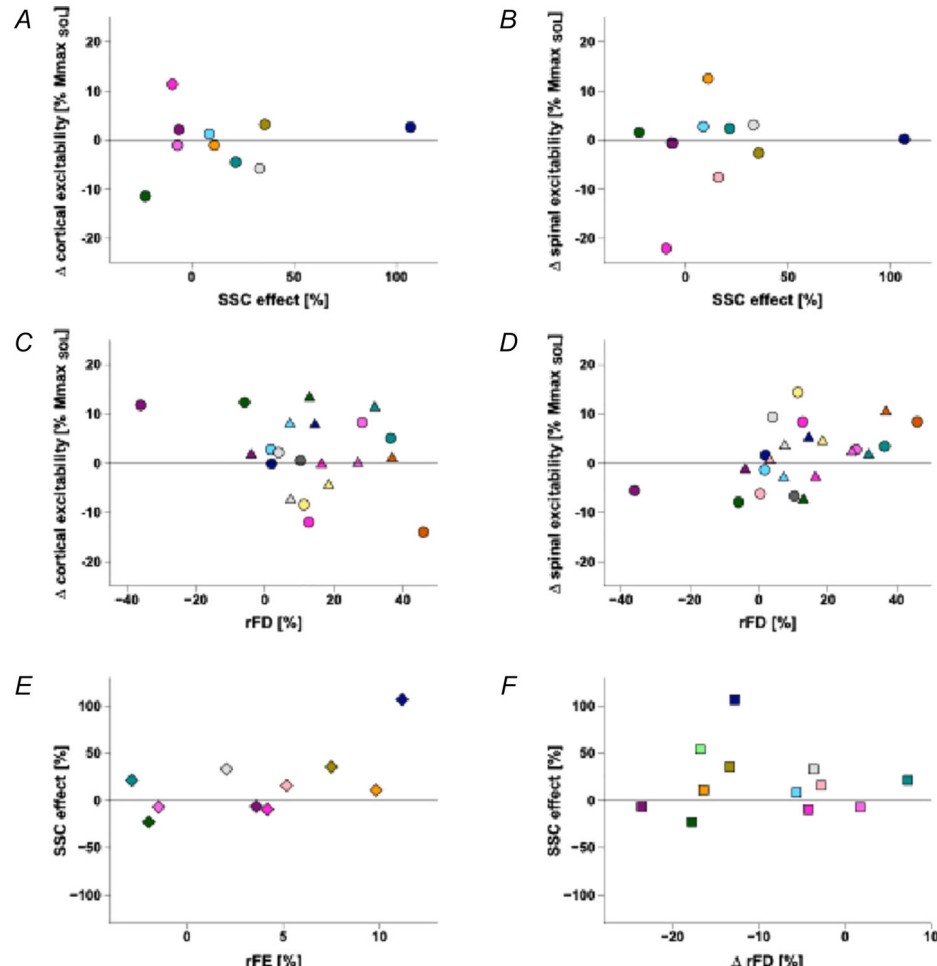

**Figure 6. Scatter plots of delta ($\triangle$) in cortical excitability (*A* and *C*) and $\triangle$ in spinal excitability (*B* and *D*) of the soleus muscle in relation to the SSC effect % (*A* and *B*) and in relation to rFD % (*C* and *D*) and the SSC effect % relative to rFE % (*E*) and rFD % (*F*)**
Pearson's correlation coefficient was calculated to assess the strength of the relations between (1) the SSC effect% or (2) between rFD% (REF$_{PF}$ *vs*. SHO, REF$_{PF}$ *vs*. SSC) and the delta cortical excitability or the delta spinal excitability. Pearson's correlation coefficient was calculated to assess the strength of the relations between the SSC effect% and (3) rFE% (REF$_{DF}$ *vs*. STR) or (4) rFD% (SHO *vs*. SSC). No significant correlations were found (*P* > 0.005). Individual data points are indicated by assigned different colours. Delta cortical excitability was calculated from the difference between normalized MEP and normalized CMEP sizes during the shortening phase (*A*) and during the steady state (*C*). Delta spinal excitability was calculated from the difference between the normalized CMEP sizes during the shortening phase (*B*) and during the steady state (*D*). Coloured circles indicate the correlation between the delta excitability and SSC effect% (SSC *vs*. SHO) (*A* and *B*) or between rFD% (REF$_{PF}$ *vs*. SSC) (*C* and *D*) and coloured triangles indicate the correlation between the delta excitability and rFD% (REF$_{PF}$ *vs*. SHO) (*C* and *D*). Coloured diamonds indicate the correlation between the SSC effect% and rFE% (*E*) and coloured squares indicate the correlation between the SSC effect% and rFD% (SHO *vs*. SSC) (*F*). [Colour figure can be viewed at wileyonlinelibrary.com]

mechanisms affect the magnitude of the SSC effect following stretch. This is why we investigated whether cortical and/or spinal excitability differed between muscle shortening when preceded by a fixed-end contraction during SHO and by active muscle stretch during SSC, respectively. Contrary to our expectations, our results showed no significant differences in normalized MEP and CMEP amplitudes during the shortening phases of SHO and SSC, which suggests that cortical and spinal excitability remained unaltered when shortening was preceded by active muscle stretch. Further, the non-significant correlations between changes in cortical and spinal excitability and the SSC effect (Fig. 6*A* and *B*) indicate that cortical and/or spinal excitability modulations did not contribute to the SSC effect. Although we did not investigate potential cortical and spinal excitability modulations during the active stretch phase, our findings indicate that potential cortical and spinal excitability modulations during active muscle stretch do not persist throughout subsequent muscle shortening during SSC. This interpretation is in contrast with the study by Hahn et al. (2012), who found larger MEPs and unchanged CMEPs in the presence of rFE during maximal voluntary plantarflexion contractions, which was interpreted as increased cortical but unchanged spinal excitability. Our findings are also contrary to Sypkes et al. (2018a), who found smaller CMEPs and an unchanged MEP-to-CMEP ratio in the presence of rFE during submaximal dorsiflexion contractions, which was interpreted as unchanged cortical, but reduced spinal excitability. A direct comparison of our results with these two studies should be made with caution given the different muscle groups and muscle lengths tested at different levels of voluntary effort. In addition, the large variability of MEPs has already been extensively discussed by Frischholz et al. (2022). Instead, by averaging 20 MEPs (Brownstein et al., 2018) in a larger sample of 14 participants, Frischholz et al. (2022) found that corticospinal excitability remained unchanged during the steady-state hold phase in the presence of rFE after active muscle stretch, which is in line with our results. In addition, the use of a monophasic pulse shape (as in Hahn et al. (2012) and Sypkes et al. (2018a)) or a biphasic pulse shape (as in Frischholz et al. (2022) and this study) to magnetically stimulate the brain can activate different neuronal populations, which favours differences in the MEPs obtained and thus in the different study results (Sommer et al., 2018).

However, if cortical and spinal excitability also remained unaltered during the stretch as previously reported in the literature (Hahn et al., 2012), then our findings indicate that the SSC effect is rather driven by mechanical than neural mechanisms, which is in line with previous interpretations (Fukutani et al., 2021).

## Cortical and spinal excitability following active muscle shortening

As a final evaluation, we investigated whether active muscle length changes led to different cortical and/or spinal excitability modulations. Therefore, we obtained MEPs and CMEPs during $REF_{PF}$ and during the steady state following shortening of SHO and SSC. As the size of the normalized MEP and CMEP responses during the steady states of $REF_{PF}$, SHO and SSC were not significantly different, contrary to our expectations our results indicate that cortical and spinal excitability remained unaltered and were not affected by any active muscle length change. Further, the non-significant correlations between changes in cortical and spinal excitability and rFD (Fig. 6*C* and *D*) indicate that cortical and/or spinal excitability modulations did not affect the steady-state torque following shortening. Contrary to our findings, Grant et al. (2017) found unchanged MEPs and unchanged CMEPs in the presence of rFD during maximal voluntary dorsiflexion contractions, which can be interpreted as absent cortical and spinal excitability modulations. In contrast, Sypkes et al. (2018b) found unchanged MEPs, but increased CMEPs in the presence of rFD during submaximal voluntary dorsiflexion contractions, which can be interpreted as decreased cortical, but increased spinal excitability. Similar to the previously mentioned studies (Hahn et al., 2012; Sypkes et al., 2018a), these studies also raise the question of whether the number of averaged MEPs and the sample size (Grant et al., 2017: MEPs = 3, $n = 13$; Sypkes et al., 2018b: MEPs = 4, $n = 11$) increase the probability of finding unreliable and overestimated effects. Therefore, more studies with larger samples and a higher number of MEPs are certainly needed to ensure that the results are not based on a type II error.

Based on our findings, we conclude that rFD was not affected by cortical and spinal excitability modulations, but that rFD is rather due to mechanisms such as cross-bridge inhibition following active muscle shortening (Chen et al., 2019; Hahn et al., 2023; Joumaa et al., 2012; Lee & Herzog, 2009; Liu et al., 2024; Raiteri et al., 2024).

## Limitations

Cortical and spinal excitability modulations were only obtained at a single instance in time around the middle phase of shortening. However, cortical and spinal excitability modulations occur rapidly (Chen & Hallett, 1999). Therefore, it is possible that cortical and spinal excitability modulations occurred earlier or later during shortening. Further, cortical and spinal excitability modulations as assessed by MEPs and CMEPs do not reveal potential specific interneuronal inhibitory and excitatory mechanisms (Kalmar, 2018). Also, our cortico-

spinal excitability modulation results are specific to our set-up, which involved targeting a specific hotspot with TMS using a figure-eight coil, using fixed stimulation parameters (Siebner et al., 2022), and having participants in a prone position. Due to time constraints, we were also unable to reach a minimum of 20 MEPs (Brownstein et al., 2018). Further, we cannot rule out differences in fascicle length affecting the torque output during the steady state of the REF, SHO and SSC conditions. At the final ankle angle of 10° PF, the plantar flexor muscles operate on the ascending limb of the torque–angle relationship (Holzer et al. 2020). Accordingly, the significant rFD following SSC (i.e. reduced torque output) should have resulted in longer fascicle lengths with the fascicles then operating further up the ascending limb compared with REF. As operating closer to the plateau results in larger force and torque capacity, the observed rFD might be underestimated. Similarly, the non-significant rFD following SHO might be explained by longer fascicle lengths compared with REF. Finally, our findings are also specific to the controlled dynamometer conditions used, which involved stretch and shortening velocities of 40° s$^{-1}$ and 120° s$^{-1}$ over an amplitude of 25°, and do not preclude cortical and spinal excitability modulations during other SSC conditions.

## Conclusion

Our study provides the first evidence that the SSC effect is not associated with cortical and spinal excitability modulations, but rather is driven by mechanical mechanisms triggered during active stretch. Other neural mechanisms, including stretch reflex activity and preactivation did not drive the SSC effect. Future research should aim to investigate specific inhibitory and excitatory mechanisms at cortical and spinal sites to improve our understanding of motor control during SSCs.

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

## Additional information

### Data availability statement

The data from this study will be made available before publication.

### Competing interests

The authors declare they have no competing interests.

### Authors contributions

L.F.R., B.J.R., T.S., W.S. and D.H. contributed to the conception of the study; L.F.R. contributed to acquisition and analysis of data; L.F.R., B.J.R. and D.H. contributed to the design of the study, interpretation of data and drafted the manuscript; all authors contributed to revising the manuscript critically for important intellectual content. All authors have read and approved the final version of this manuscript and agree to be accountable for all aspects of the work in ensuring that questions related to the accuracy or integrity of any part of the work are appropriately investigated and resolved. All persons designated as authors qualify for authorship, and all those who qualify for authorship are listed.

## Funding

This study was supported by the German Research Foundation (DFG; HA 5977/5-1,2; SE 2109/2-1,2 and SI 841/15-1,2; project number: 354863464).

## Keywords

force depression, force enhancement, performance enhancement, stretch reflex, triceps surae

## Supporting information

Additional supporting information can be found online in the Supporting Information section at the end of the HTML view of the article. Supporting information files available:

**Peer Review History**

