## [Peer Review History · The Journal of Physiology]

The stretch-shortening cycle effect is not associated with cortical or spinal excitability modulations

Lea-Fedia Rissmann, Daniel Hahn, Brent James Raiteri, Tobias Siebert, and Wolfgang Seiberl
DOI: 10.1113/JP287508

Corresponding author(s): Daniel Hahn (daniel.hahn@rub.de)

Review Timeline:

Submission Date:	16-Aug-2024
Editorial Decision:	27-Sep-2024
Revision Received:	15-Apr-2025
Accepted:	30-Apr-2025

Senior Editor: Richard Carson

Reviewing Editor: Ross Pollock

Transaction Report:

Dear Dr Rissmann,

Re: JP-RP-2024-287508 "The stretch-shortening cycle effect is not associated with cortical or spinal excitability modulations" by Lea-Fedia Rissmann, Daniel Hahn, Brent James Raiteri, Tobias Siebert, and Wolfgang Seiberl

Thank you for submitting your manuscript to The Journal of Physiology. It has been assessed by a Reviewing Editor and by 2 expert referees and we are pleased to tell you that it is potentially acceptable for publication following satisfactory major revision.

REVISION CHECKLIST:

Please upload two versions of your manuscript text: one with all relevant changes highlighted and one clean version with no

changes tracked. The manuscript file should include all tables and figure legends, but each figure/graph should be uploaded as separate, high-resolution files.

We look forward to receiving your revised submission.

Yours sincerely,

Richard Carson
Senior Editor
The Journal of Physiology

REQUIRED ITEMS FOR REVISION

- Author photo and profile. First or joint first authors are asked to provide a short biography (no more than 100 words for one author or 150 words in total for joint first authors) and a portrait photograph. These should be uploaded and clearly labelled together in a Word document with the revised version of the manuscript. See Information for Authors for further details.

- Please upload separate high-quality figure files via the submission form.

- Papers must comply with the Statistics Policy: https://jp.msubmit.net/cgi-bin/main.plex?form_type=display_requirements#statistics.

In summary:

- If n {less than or equal to} 30, all data points must be plotted in the figure in a way that reveals their range and distribution. A bar graph with data points overlaid, a box and whisker plot or a violin plot (preferably with data points included) are acceptable formats.
- If $n > 30$, then the entire raw dataset must be made available either as supporting information, or hosted on a not-for-profit repository, e.g. FigShare, with access details provided in the manuscript.
- 'n' clearly defined (e.g. x cells from y slices in z animals) in the Methods. Authors should be mindful of pseudoreplication.
- All relevant 'n' values must be clearly stated in the main text, figures and tables.
- The most appropriate summary statistic (e.g. mean or median and standard deviation) must be used. Standard Error of the Mean (SEM) alone is not permitted.
- Exact p values must be stated. Authors must not use 'greater than' or 'less than'. Exact p values must be stated to three significant figures even when 'no statistical significance' is claimed.

- Please include an Abstract Figure file, as well as the Figure Legend text within the main article file. The Abstract Figure is a piece of artwork designed to give readers an immediate understanding of the research and should summarise the main conclusions. If possible, the image should be easily 'readable' from left to right or top to bottom. It should show the physiological relevance of the manuscript so readers can assess the importance and content of its findings. Abstract Figures should not merely recapitulate other figures in the manuscript. Please try to keep the diagram as simple as possible and without superfluous information that may distract from the main conclusion(s). Abstract Figures must be provided by authors no later than the revised manuscript stage and should be uploaded as a separate file during online submission labelled as File Type 'Abstract Figure'. Please also ensure that you include the figure legend in the main article file. All Abstract Figures should be created using BioRender. Authors should use The Journal's premium BioRender account to export high-resolution

images. Details on how to use and access the premium account are included as part of this email.

- Please include a full title page as part of your main article (Word) file, which should contain the following: title, authors, affiliations, corresponding author name and contact details, keywords, and running title.

- The corresponding author must provide an institutional email address (not a personal address) for their author account. We encourage ALL co-authors to also provide institutional email addresses. If this cannot be provided (as corresponding author), then a stamped letter must be provided from the institution which confirms their role and employment there (please upload this with the revised submission).

EDITOR COMMENTS

Reviewing Editor:

Thank you for submitting your manuscript to the Journal of Physiology. It has been reviewed by two independent experts who have raised some points that should be considered to aid the impact of the manuscript. In particular you should consider shortening the length of the introduction and discussion to make it more concise, while doing this consideration should be given to the presentation of the aims and hypothesis in light of the complex experimental design and number of conditions tested (which is a strength of the manuscript). Also, it was noted that some of the findings were not in line with previous work published by your group - greater consideration needs to be given as to why this may be the case given that the work was conducted in the same laboratory.

Please ensure that the tables and/or the table legends indicate the number of participants tested.

Senior Editor:

The authors have adopted a null-hypothesis significance testing (NHST) approach (as reflected in their choice of statistical analyses). This being the case, statements such as "non-significantly larger" have the potential to mislead. In circumstances in which "statistical significance" is not achieved (i.e. given adoption of NHST) the criterion adopted by the authors (i.e., in relation to the significance level set a priori) has not been met, and terms such as "larger" or "smaller" should not be used. In the event that the authors wish to refer to such contrasts in another way, please consider reporting effect size statistics, along with the associated confidence intervals.

REFEREE COMMENTS

Referee #1:

I reviewed the paper "The stretch-shortening cycle effect is not associated with cortical or spinal excitability modulations." The paper considers the role of neural excitability to residual force depression and stretch-shortening cycle properties in vivo. The experiment is timely and is of interest to a large portion of the muscle field, who are interested in the history-dependence of joint torque and muscle tension during cyclical activities like jumping and running. There are mainly no studies on this topic, so these are certainly new results. The experiments are well conducted and use current state-of-the-art approaches to track changes to cortical and spinal excitability under different conditions. They find little evidence that they account for rFD and SSC changes (at least under these conditions). The intro and discussion are a bit long for the amount of data they collect, so I would suggest tightening it up. I also provide a couple strategies to do this below. I also think the figures (7 in total), could be combined to ~4. Finally, I provide some comments that are in the spirit of helping the interested but novice reader to follow along with the important points of the story.

Overall, I think that this paper is well suited for the Journal of Physiology and after revisions, could be acceptable for publication and be a nice addition to the literature.

Introduction

1) 56-60: Make clear that the muscle is contracting during the entire SSC, not just shortening. This is a needed detail for the novice.

2) 64: The term "tFE" has been used by others, so remove "which we have"

3) 65: "Could" should be exchanged for "does"

4) Somewhere in the second paragraph, it should be indicated that SSC effects occur even at the single fiber or myofibril level, demonstrating that at least a part of RFE and SSC effect is caused at the level of the sarcomere. Another component for in vivo experiments are neural related, which is a logical consideration for sure.

5) Line 73: Instead of "however", exchange for "In addition to sub-cellular processes, neural mechanisms....". This is more clear.

6) Line 79-81. Make clear to the novice that this situation would reduce force, opposite of that needed to explain SSC.

7) 91: "which usually was attributed to...". Maybe just "which was attributed to..." because the reader will wonder what the other attributed pieces would be.

8) 102: I think worthwhile to indicate MEPs are very variable and thus higher sample sized are warranted.

9) The introduction certainly has all the details needed to explain the field. I think that there are more examples than needed to get the reader on board that spinal and cortical excitability could answer some of the SSC properties. I think it is safe to more broadly discuss how these excitabilities can modulate in vivo muscle force, and select just a couple top examples from the literature to drive home the point. And then, save the rest for the discussion (which you do mainly bring up anyhow). The intro would be a bit tighter and I think keep more readers engaged and not checked out from so much background. I leave it to the authors to address this comment as they see fit (also with other reviewer's comments).

Methods

1) Participant exclusion / inclusion criteria are well addressed.

2) 145-159: I would add to the representation of the setup and subject position the straps that are indicated in the text. I would also refer to the cartoon of the setup earlier in the paragraph, as those who do not use dynamometers will quickly get confused. Further, previous papers that used similar setups could be referred to for the interested reader.

3) Figure 1: I would consider having some wires going to from the electrodes and such on the subject, to appropriate labeled computer boxes, to help the novice know what is stimulating and what is recording. I do understand the figure will get a bit busy, so it can be simple additions.

4) Methods are overall well described. If space considerations are needed, a lot of this can go to the SI or be cited out. But best to leave it all here for paper completeness.

Results

1) Any stats done in Tables 1 and 3? Should be indicated in the table legends. In all tables, please indicate that stats tests used. Further, many journal now require putting the n-values in all these as well, which I think could be put in parentheses after the stated mean +/- SD.

2) In the results, please indicate when observed changes are "as expected" or "as previously described" and when something unusual happens. This will help the novice reader keep track of the new news generated from the work. The discussion is where these things will be explored, but giving the reader at least some indication of what the interesting things are in the results is worthwhile.

3) 378: Was the data with stretch reflex treated differently?

4) It is good that the correlation analysis provides 95% CIs!

5) Figures: I overall like the figures and designs are clean. I do think that some of them can be combined, and panels made a bit smaller, to conserve space and make 4 real nice figures. 7 figures is a lot for the amount of data presented. I think 2 and 3 can go together. Figures 4 and 5 could also be put together, with a little extra labelling to keep clear what is what. 6 and 7 can also go together. I also understand why these would be kept separated, so and changes in the spirit of this comment (less figures) would be OK.

6) Figure legends: Used stats tests in legends are needed, even if things turned out to be NS.

Discussion

- 1) 397-398: May be strongest to lead the discussion with this sentence. Or maybe as the second sentence. Lets the reader know the exciting news right from the start.
- 2) 401-411: Some may be wondering if this was all tendon stretch and little fiber/fascicle stretch. Could the authors refer to an ultrasound study that indicates this joint rotation was enough to elicit significant fascicle length changes? I am not 100% sure something like this is published. But anything to provide some relief to this potential issue would be helpful.
- 3) 417: Obviously not apples to apples, but how does the SSC effect magnitude compare to those done at the Fiber or myofibril level?
- 4) 426: Several papers by Herzog / Fukutani / Nishikawa groups also looked at this. A citation is warranted I think.
- 5) 429: "history-dependent".
- 6) 434-436: If providing non-significant means, I suppose the author thinks them still trending and thus useful to consider. In this case, I would still put the P-values, and/or direct the reader to the figure to explain why there is evidence for trending and therefore reporting the means.
- 7) 440: The "taken together" sentence sounds like the last sentence of a paragraph, so consider breaking this paragraph up. The sentence is also confusing to me. How can the rFD mechanism alone account for the SSC and RFD, if the previous stretch was not affecting it? Or maybe the sentence does not yet convey the message the authors intend. Please rephrase.
- 8) 452-462: So the stretch reflex is not mandatory for the history-dependent properties, but may enhance them under specific conditions? If so, this should be stated plainly.
- 9) 465-66 - sentence is awkward. Please rephrase.
- 10) 486: "As a final evaluation, we investigated whether..." is clearer.
- 11) First sentence of limitations not needed.
- 12) One limitation that often arises, and I discussed a bit above, is that we are never really certain about the amount of stretching and shortening occurring at the fascicle level unless we are directly tracking it with ultrasound. Based on MTU dynamics that these authors are well acquainted with, there is a good chance that the final fascicle lengths will be different between the reference state, rFD and SSC states, and may account for some of the torque (and neural?) results. It is important to explain and explore this consideration in the methods, either as a paragraph in the limitations, or in another way throughout the discussion. It may even be possible to estimate differences in fascicle lengths, and what this would mean for differences in torque. I should point out that I do not think this is a fatal flaw of the method, as it is hard to do every single evaluation on the muscles at the same time. Further, I believe most of the papers cited for comparison also did experiments similarly, so the data is comparable to those in the field. All that being said, there is a portion of the muscle community who will disregard this research solely on this point, so it is important to attack it head on. Hence why I ask for the additions.

Referee #2:

"The stretch-shortening cycle effect is not associated with cortical or spinal excitability modulations" study sought to investigate the contributions of cortical and spinal excitability to force production in the stretch shortening cycle (SSC) relative to a pure shortening (SHO) contractions. The overall conclusion was the SSC is neither associated with cortical or spinal excitability. Although the paper is well-written the challenge is the complexity of using 5 different conditions to address the identified hypothesis of greater torque output from SSC compared with shortening (SHO) alone. The reader has to 'hunt' and somewhat assume the necessity for all conditions. Although Fig 3 is nicely laid out, and the electrical stimulation and protocol described the benefit of multiple conditions to address a comparative hypothesis should be strengthened, or the hypotheses better representative of the experimental design to strengthen the impact of this research.

Authors from this study have previously reported (for example Hahn et al. 2012) larger motor evoked potentials (MEPs) in the presence of rFE during maximal voluntary plantar flexion contractions, which was interpreted as increased cortical, but unchanged spinal excitability (lines 95-98). This is an example of where some prior work is not aligning with findings from this study. This challenges the understanding of the physiological mechanisms. Addressing discrepancies directly and ensuring that all hypotheses are clearly addressed in the Discussion and in more detail (see small discussion lines 481-483 and Hyp 3 around long-lasting effects) would strengthen the paper.

It is difficult to understand where the Independent Variable of stimulation was tested. This might arise from confusion between using the linear mixed-effect models and ANOVAs (eg., two-way ANOVA; contraction condition x muscle). Line 361 identifies no effect of the stimulation technique for what seems to be an ANOVA. Identify and justify why linear mixed-effect models as well as ANOVAs were used, and how the stimulation conditions were tested (they are identified in the regressions). The statistics also fail to identify if the data was normally distributed, and thus the appropriacy of a parametric design cannot be assessed. The number of data points in the regressions are highly variable, explain how the data was included and/or excluded and how this influences the power of the statistics.

Line 126: define SHO on first use

Line 134: likely insufficient data to consider sex-differences. However, have two different shapes for females and males in figures and/or discuss potential, through observation, how sex MIGHT be influencing the findings.

Line 210: clarify hotspot marking on skin, notably in females where this can be problematic for repeatability.

Line 231: the 't' is missing in sTimulation

Line 286-288: It seems like these are the minimum number of potentials that are included in the mean data. Report the average and/or range of what was included in the Results.

Line 393-394: Be specific on the significant effect found

Line 403: Consider rephrasing, as "We have termed this as..." makes it seem like this is the first use of these terms. Provide references.

Line 404: Incomplete sentence

Figures: consider using shapes and/or colors more systematically and in so doing identify the sex of the participants to increase robust reporting and contribute to understanding sex-differences in these contractions.

END OF COMMENTS

Reviewing Editor:

Thank you for submitting your manuscript to the Journal of Physiology. It has been reviewed by independent experts who have raised some points that should be considered to aid the impact of the manuscript. In particular you should consider shortening the length of the introduction and discussion to make it more concise, while doing this consideration should be given to the presentation of the aims and hypothesis in light of the complex experimental design and number of conditions tested (which is a strength of the manuscript). Also, it was noted that some of the findings were not in line with previous work published by your group - greater consideration needs to be given as to why this may be the case given that the work was conducted in the same laboratory.

Please ensure that the tables and/or the table legends indicate the number of participants tested.

Senior Editor:

The authors have adopted a null-hypothesis significance testing (NHST) approach (as reflected in their choice of statistical analyses). This being the case, statements such as "non-significantly larger" have the potential to mislead. In circumstances in which "statistical significance" is not achieved (i.e. given adoption of NHST) the criterion adopted by the authors (i.e., in relation to the significance level set a priori) has not been met, and terms such as "larger" or "smaller" should not be used. In the event that the authors wish to refer to such contrasts in another way, please consider reporting effect size statistics, along with the associated confidence intervals.

We thank the reviewing editor and the senior editor for evaluating our manuscript and for inviting experts in neuromuscular function to review our paper. We also thank them for the chance to resubmit and for recognizing our study's strengths.

We now shortened the introduction, while keeping the length of the discussion similar in order to address all important points. Also, as suggested by the two referees, we have now presented our aims and hypotheses more clearly and made it easier for the reader to understand why the experimental protocol was necessary to address our hypotheses (lines 123-131). As the referees noted that some of our results are not consistent with our previous work, this has now also been addressed in more detail (lines 507-516). Finally, we ensured that the number of participants tested is indicated in the table legends (Tab. 1 to 3).

Based on the senior editor's comment, we decided to not use terms such as "larger" or "smaller" in cases where statistical significance was not achieved. Instead, we now use terms such as "similar" or "not significantly different".

Please find our point-by-point responses to referees' comments below. The reviewer comments are in black and our responses are highlighted in green.

Referee #1:

We thank referee #1 for their effort in reviewing our manuscript and for their constructive criticism. Please find our point-by-point responses to your concerns below.

Introduction

1) 56-60: Make clear that the muscle is contracting during the entire SSC, not just shortening. This is a needed detail for the novice.

Thank you for pointing out that it was not clear that the stretch phase is also performed actively. We now specify that muscles actively stretch and actively shorten during SSCs. This should make it clear to the reader that the muscle contracts during the entire SSC.

“During everyday locomotion, lower limb muscle-tendon units (MTU) often actively stretch before actively shortening, which is known as a stretch-shortening cycle (SSC).”

Line 56-57

2) 64: The term "tFE" has been used by others, so remove "which we have"

We agree and removed this manuscript text as suggested by the referee.

Line 65-66

3) 65: "Could" should be exchanged for "does"

We disagree that „could“ should be replaced with “does” because we are talking about the forces during shortening after stretch, not the forces during stretch. Consequently, tFE would need to be present at the end of stretch for forces to be higher at the start of shortening. If tFE is present during stretch, but not at the end of stretch, tFE would not lead to higher forces at the start of shortening. Additionally, tFE at the end of stretch would not necessarily lead to higher forces throughout shortening. This is why we created a conditional sentence here using „could“ and we would prefer to keep it this way.

4) Somewhere in the second paragraph, it should be indicated that SSC effects occur even at the single fiber or myofibril level, demonstrating that at least a part of RFE and SSC effect is caused at the level of the sarcomere. Another component for in vivo experiments are neural related, which is a logical consideration for sure.

Thank you for pointing this out. We now state:

“As the SSC effect has been observed at the single-fibre level, mechanical factors are at least partly located in the sarcomeres:[...]”

Line 62-64

5) Line 73: Instead of "however", exchange for "In addition to sub-cellular processes, neural mechanisms....". This is more clear.

We accepted the referee's suggestion and changed the manuscript text accordingly as follows:

"In addition to mechanical factors at the sarcomeric level neural mechanisms [...]"

Line 76

6) Line 79-81. Make clear to the novice that this situation would reduce force, opposite of that needed to explain SSC.

We agree with the comment and realize that this would not be immediately clear to the novice reader. We now state:

"In contrast, the SSC effect might be attenuated by inhibitory mechanisms that are triggered during active muscle stretch, reducing forces during successive shortening, opposing the SSC effect [...]"

Line 82-84

7) 91: "which usually was attributed to...". Maybe just "which was attributed to..." because the reader will wonder what the other attributed pieces would be.

Changed as suggested.

8) 102: I think worthwhile to indicate MEPs are very variable and thus higher sample sized are warranted.

We agree with this comment and think that it is worthwhile to indicate that MEPs are very variable and thus a higher sample size is recommended. We now mention more precisely the number of required MEPs in the discussion.

"[...] In addition, the large variability of MEPs has already been extensively discussed by Frischholz et al., (2022). Instead, by averaging 20 MEPs (Brownstein et al., 2018) in a larger sample of 14 participants, Frischholz et al. (2022) found [...]"

Line 507-516

9) The introduction certainly has all the details needed to explain the field. I think that there are more examples than needed to get the reader on board that spinal and cortical excitability could answer some of the SSC properties. I think it is safe to more broadly discuss how these excitabilities can modulate in vivo muscle force, and select just a couple top examples from the literature to drive home the point. And then, save the rest for the discussion (which you do mainly bring up anyhow). The intro would be a bit tighter and I think keep more readers engaged and not checked out from so much background. I leave it to the authors to address this comment as they see fit (also with other referee's comments).

Thank you for this feedback. We tried to discuss the most important examples regarding cortical and spinal excitability modulation within the lower extremities to make the introduction a tighter. We hope that this will keep the reader more engaged (see lines 100-103).

Methods

1) Participant exclusion / inclusion criteria are well addressed.

Thank you for acknowledging this.

2) 145-159: I would add to the representation of the setup and subject position the straps that are indicated in the text. I would also refer to the cartoon of the setup earlier in the paragraph, as those who do not use dynamometers will quickly get confused. Further, previous papers that used similar setups could be referred to for the interested reader.

As suggested by the referee, we have now made the hip belt visible in the figure and hope that this now provides a better visual representation of the set-up. We also agree with the referee that the set up can be mentioned earlier to avoid confusion and now refer to the figure in lines 147-148. Further, we now refer to Frischholz et al. (2022) who used a similar set up.

“Resultant ankle joint torque and crank arm angle were recorded from the right foot of the participants’ leg, while participants laid prone on the bench of a dynamometer (IsoMed2000, Ferstl GmbH, GER) (Fig. 1A). The right foot was tightly strapped onto a footplate attachment to minimize heel lift during contractions (Fig. 1B). The hip and knee joints remained fully extended, and accessory movements during contractions were minimized by securing the participants’ waist with a belt to the dynamometer bench (Firschholz et al., 2022).”

Line 145- 150

3) Figure 1: I would consider having some wires going to from the electrodes and such on the subject, to appropriate labeled computer boxes, to help the novice know what is stimulating and what is recording. I do understand the figure will get a bit busy, so it can be simple additions.

We understand the referee’s aim to make things as clear as possible for novices, however, we think that additional cables and computer boxes would overcrowd the figure and potentially distract the reader from the figure’s main aim of showing the overall setup. Accordingly, we’d like to abstain from adding equipment, but we decided to add a voltage symbol next to the

stimulation electrodes. We believe that this will help the novice reader to understand which electrodes are stimulating and which are recording. This is also indicated in the figure's legend now. The figure legend now reads:

“Electrodes for spinal cord stimulation (voltage symbol) [...]. Electrode placement of the cathode for electrical nerve stimulation (voltage symbol) [...].”

Fig. 1

4) Methods are overall well described. If space considerations are needed, a lot of this can go to the SI or be cited out. But best to leave it all here for paper completeness.

Thank you for acknowledging this. We further agree with the referee that it is best to leave all methods within the paper for completeness. Therefore, we did not move information to the SI and/or cite things out.

Results

1) Any stats done in Tables 1 and 3? Should be indicated in the table legends. In all tables, please indicate that stats tests used. Further, many journal now require putting the n-values in all these as well, which I think could be put in parentheses after the stated mean +/- SD.

We thank the referee for their suggestion and now provide the requested information in the table legends.

Table 1:

“We used a two-way repeated-measures restricted maximum likelihood mixed-effects model to assess mean differences in normalized EMG amplitudes. The mean normalized EMG_{SOL}, EMG_{MG}, EMG_{LG}, and EMG_{TA} were not significantly affected by contraction condition. Mean ± SD values are presented in % (n = 15).”

Table 2:

“We used a paired t-test (during shortening: SHO vs. SSC), a one-way repeated-measures ANOVA (during steady state: REF_{PF} vs. SHO vs. SSC) and a Wilcoxon test (during steady state: REF_{DF} vs. STR) to assess mean differences in resultant plantar flexion torque. Mean ± SD values are presented in Nm (n = 12). [...]”

Table 3:

“We used a paired t-test (during shortening: SHO vs. SSC) and a one-way repeated-measures ANOVA (during steady state: REF_{PF} vs. SHO vs. SSC) to assess mean differences in M_{max} among contraction conditions. Following a significant interaction, we conducted Holm-Šídák multiple comparisons. Further, a two-way repeated-measures restricted maximum likelihood mixed-effects model was used to assess mean differences in normalized CMEP and MEP responses. There were no significant differences found. Mean ± SD values of the maximum M-wave (Mmax) size (n = 15), the cervicomedullary motor-evoked potential (CMEP) size (n = 13 during shortening, n = 12 during steady state), and the motor-evoked potential (MEP) size (n = 14 during shortening, n = 13 during steady state). [...]”

2) In the results, please indicate when observed changes are "as expected" or "as previously described" and when something unusual happens. This will help the novice reader keep track of the new news generated from the work. The discussion is where these things will be explored, but giving the reader at least some indication of what the interesting things are in the results is worthwhile.

We have revised the manuscript and see the referee's point as helpful feedback. However, we decided not to indicate in the results when the observed changes are "as expected" or "as previously described" and when something unusual happens. We now categorize this more specifically in the discussion.

"As expected, we found significant tFE of 15 ± 25 % at the end of stretch and significant rFE of 7 ± 3 %. following stretch."

Line 421- 422

"As expected, we found that the mean torque during the shortening phase of SSC was significantly higher compared with SHO [...]"

Line 433-434

"[...] torque following SHO was similar compared with REF_{PF} and significantly larger by ~8% than the mean torque following SSC ($p = 0.011$), (Fig. 3C, Fig. 4B). This is contrary to our expectations and to previous literature reporting rFE, non-significant rFD or reduced rFD following SSCs (Fortuna et al., 2017; Fortuna et al., 2018; Hahn et al., 2018)."

Line 455-459

"[...] SLR stretch reflexes were found in only one out of fifteen participants for SOL, but not for MG or LG. This is in contrast to studies reporting a significant contribution to the SSC effect by stretch-reflex activity [...]"

Line 475-477

"Contrary to our expectations, our results showed no significant differences in normalized MEP and CMEP amplitudes during the shortening phases of SHO and SSC, which suggests that cortical and spinal excitability remained unaltered when shortening was preceded by active muscle stretch."

Line 491-494

"[...] contrary to our expectations our results indicate that cortical and spinal excitability remained unaltered and were not affected by any active muscle length change."

Line 527-529

"[...] our findings indicate that the SSC effect is rather driven by mechanical than neural mechanisms, which is in line with previous interpretations (Fukutani et al., 2021)."

Line 518-520

“Contrary to our findings, Grant et al. (2017) found unchanged MEPs and unchanged CMEPs in the presence of rFD [...]. In contrast, Sypkes et al. (2018b) found unchanged MEPs, but increased CMEPs in the presence of rFD [...].”

Line 531-533

3) 378: Was the data with stretch reflex treated differently?

Yes, the stretch-reflex data was treated differently. As only one out of 15 participants showed stretch-reflex activity, and only in the soleus muscle, we did not perform a statistical analysis. We have now explicitly mentioned this in the statistics section:

“No statistical analysis was performed on the stretch-reflex data, as stretch reflexes were detected in one of 15 participants only.”

Line 358-359

4) It is good that the correlation analysis provides 95% CIs!

Thank you for the positive feedback.

5) Figures: I overall like the figures and designs are clean. I do think that some of them can be combined, and panels made a bit smaller, to conserve space and make 4 real nice figures. 7 figures is a lot for the amount of data presented. I think 2 and 3 can go together. Figures 4 and 5 could also be put together, with a little extra labelling to keep clear what is what. 6 and 7 can also go together. I also understand why these would be kept separated, so and changes in the spirit of this comment (less figures) would be OK.

We are happy that the referee likes our figures. We also understand the referee's comment and we agree that it would certainly be space-saving to combine the figures as suggested. However, we decided against combining figures 2 & 3 because figure 2 contains data from one participant only, whereas figure 3 contains data from the participant group. Additionally, adding figure 2's panel to figure 3 creates an uneven number of panels that is visually unappealing. We also did not combine figures 4 and 5 because the outcome variable was different (torque versus stimulation response). However, we combined figures 6 & 7 as they match in terms of content (correlations), the panels only have one column each, and we added an additional panel to result in an even number of panels, which is visually appealing. The new figure 6 is provided below.

6) Figure legends: Used stats tests in legends are needed, even if things turned out to be NS.

As requested by the referee, we now include the statistical tests in the legend of Figures 4, 5 & 6.

Figure 4:

“We used a paired t-test to assess mean differences in averaged torque between SHO and SSC during shortening (A), a one-way repeated-measures ANOVA to assess mean differences in averaged torque between REF_{PF}, SHO and SSC following shortening (B) and a Wilcoxon test to assess mean differences in torque between REF_{DF} and STR (C). Torque during the shortening phase was significantly enhanced during SSC compared with SHO (A), significantly lower following SSC compared with REF_{PF} (B) and compared with SHO (C) and significantly enhanced following STR compared with REF_{DF}. “

Figure 5:

“Neither a paired t-test (during shortening: SHO vs. SSC) nor a one-way repeated-measures ANOVA (during steady state: REF_{PF} vs. SHO vs. SSC) indicated significant differences in mean M_{max} among contraction conditions (A, B). Further a two-way repeated-measures restricted

maximum likelihood mixed-effects model did not indicate mean differences in normalized MEP (C, D) or CMEP responses (E, F)."

Figure 6:

"Pearson correlation coefficients were calculated to assess the strength of the relations between (1) the SSC effect% or (2) between rFD% (REF_{PF} vs. SHO, REF_{PF} vs. SSC) and the delta cortical excitability or the delta spinal excitability. Pearson correlation coefficients were also calculated to assess the strength of the relations between the SSC effect% and (3) rFE% (REF_{DF} vs. STR) or (4) rFD% (SHO vs. SSC). No significant correlations were found ($p \geq 0.005$)."

Discussion

1) 397-398: May be strongest to lead the discussion with this sentence. Or maybe as the second sentence. Let the reader know the exciting news right from the start.

We thank the referee for their comment and now start the discussion with the sentence suggested by the referee.

"Our results indicate that the SSC effect is unlikely to be associated with cortical or spinal excitability modulations."

Lines 408-409

2) 401-411: Some may be wondering if this was all tendon stretch and little fiber/fascicle stretch. Could the authors refer to an ultrasound study that indicates this joint rotation was enough to elicit significant fascicle length changes? I am not 100% sure something like this is published. But anything to provide some relief to this potential issue would be helpful.

The referee raises an important point. We collected pilot data on N=1 to show that the soleus muscle-tendon unit stretch caused by an imposed ankle dorsiflexion actually led to fascicle stretch of ~10 mm within the soleus (see Fig. R1 below). Additionally, a recent publication showed that a joint rotation from 10° plantar flexion to 15° dorsiflexion at an angular velocity of 40°·s⁻¹ during submaximal plantar flexion contractions of 30-40% MVC resulted in fascicle stretch of 12±3 mm within the medial gastrocnemius (Goeking et al., 2024, Holzer et al., 2024). As we used an identical preload and joint rotation during our SSC condition, it is reasonable to expect that the soleus muscle was also stretched considerably during soleus muscle-tendon unit stretch, which is supported by our pilot results. We have made the following changes to our manuscript text to indicate this:

"To ensure that our range of motion and selected pre-activation was sufficient to induce fascicle stretch in the soleus muscle, we collected ultrasound pilot data on N=1 and observed soleus fascicle stretch of ~10 mm."

Lines 425-428

Figure R1. Change in soleus fascicle length during the SSC condition in one participant during pilot testing.

3) 417: Obviously not apples to apples, but how does the SSC effect magnitude compare to those done at the Fiber or myofibril level?

In literature, SSC effects of up to 30% were reported when testing single skinned fibers (Fukutani et al., 2017, Fukutani and Herzog, 2019, Tomalka 2020, 2021). Thus, the SSC effect magnitude appears to be larger at the single skinned fiber level. However, and as pointed out by the referee, this is not comparing apples to apples as the structural integrity of skinned fibers is compromised, affecting lattice spacing, which in turn might result in different cross-bridge kinetics during activation and movement compared with muscle fibers contracting *in vivo*.

Nevertheless, we addressed the referee's comment and now briefly mention the in-vitro SSC effect magnitude (Lines 444-447):

"[...] the SSC effect is partly due to contractile components, which recently was further supported by work that found SSC effects of up to 30% using skinned fiber preparations (Fukutani et al., 2017, Fukutani and Herzog, 2019, Tomalka et al. 2020, 2021)."

4) 426: Several papers by Herzog / Fukutani / Nishikawa groups also looked at this. A citation is warranted, I think.

We thank the referee for pointing this out and added the mentioned studies.

"[...]the SSC effect is partly due to contractile components, which recently was further supported by work that found a SSC effect of up to 30% using skinned fiber preparations (Fukutani et al., 2017, Fukutani and Herzog, 2019, Tomalka et al. 2020, 2021)."

Line 444-447

5) 429: "history-dependent".

We changed the title to "History-dependent muscle forces".

Lines 449

6) 434-436: If providing non-significant means, I suppose the author thinks them still trending and thus useful to consider. In this case, I would still put the P-values, and/or direct the reader to the figure to explain why there is evidence for trending and therefore reporting the means.

We strongly disagree with interpreting "trending" results as significant because this is misleading as a larger sample size might not lead to significant findings. Therefore, we have replaced "non-significantly reduced" with "similar".

Lines 453-456

"Although we observed rFE following active muscle stretch, we found a significant rFD of ~13% following SSC ($p = 0.006$), while the mean torque following SHO was similar compared with REF_{PF}, and significantly larger by ~8% than the mean torque following SSC ($p = 0.011$), (Fig. 3C, Fig. 4B)."

7) 440: The "taken together" sentence sounds like the last sentence of a paragraph, so consider breaking this paragraph up. The sentence is also confusing to me. How can the rFD mechanism alone account for the SSC and RFD, if the previous stretch was not affecting it? Or maybe the sentence does not yet convey the message the authors intend. Please rephrase.

Thank you for pointing out the unclear meaning of this sentence. We modified and broke up the sentence into two sentences and we added a correlation between the SSC effect and rFE (non-significant). It now reads as:

"Further, our results showed no significant correlations between the SSC effect and the magnitudes of rFE and rFD, respectively (Fig. 6E & F). These results suggest that under the given contraction conditions, the stretch-induced mechanisms underpinning rFE did not contribute much or at all to the observed SSC effect. Further, the larger rFD following SSC and the similar torque following SHO compared with REF_{PF} suggests that the mechanisms underpinning rFD rather attenuated the SSC effect."

Lines 459 - 464

We hope that any confusion is resolved and that our message is clear now.

8) 452-462: So the stretch reflex is not mandatory for the history-dependent properties, but may enhance them under specific conditions? If so, this should be stated plainly.

Thank you for pointing this out. We added the following sentence:

"Accordingly, stretch reflex activity could possibly contribute to the SSC effect in everyday"

movements, but this was not the case for our dynamometer-specific contraction conditions.”
Line 481-483

9) 465-66 - sentence is awkward. Please rephrase.

According to the referee's comment, we rephrased the sentence:

“As rFE has been linked with modulations in cortical and spinal excitability (Gruber et al., 2009; Hahn et al., 2012; Sypkes et al., 2018a), it is also possible that neural mechanisms affect the magnitude of the SSC effect following stretch. This is why we investigated...”
Lines 486-488

10) 486: "As a final evaluation, we investigated whether..." is clearer.

Thank you for your suggestion, which we incorporated in the text:

“As a final evaluation, we investigated whether active muscle length changes led to different cortical and/or spinal excitability modulations. Therefore, we obtained...”
Lines 523-524

11) First sentence of limitations not needed.

Deleted as suggested.

12) One limitation that often arises, and I discussed a bit above, is that we are never really certain about the amount of stretching and shortening occurring at the fascicle level unless we are directly tracking it with ultrasound. Based on MTU dynamics that these authors are well acquainted with, there is a good chance that the final fascicle lengths will be different between the reference state, rFD and SSC states, and may account for some of the torque (and neural?) results. It is important to explain and explore this consideration in the methods, either as a paragraph in the limitations, or in another way throughout the discussion. It may even be possible to estimate differences in fascicle lengths, and what this would mean for differences in torque. I should point out that I do not think this is a fatal flaw of the method, as it is hard to do every single evaluation on the muscles at the same time. Further, I believe most of the papers cited for comparison also did experiments similarly, so the data is comparable to those in the field. All that being said, there is a portion of the muscle community who will disregard this research solely on this point, so it is important to attack it head on. Hence why I ask for the additions.

The referee is right that we would have needed ultrasound imaging to be able to quantify the amount of fascicle stretch and shortening in the triceps surae during our experimental conditions. Accordingly, soleus's final fascicle lengths determined from ultrasound images might have been significantly different between the reference, rFD, and SSC states. Consequently, differences in fascicle length may help to explain the torque results, which we have now discussed as a limitation of our study:

“Further, we cannot rule out differences in fascicle length affecting the torque output during the steady state of the REF, SHO, and SSC conditions. At the final ankle angle of 10° plantar flexion, the plantar flexor muscles operate on the ascending limb of the torque-angle relationship (Holzer et al. 2020). Accordingly, the significant rFD following SSC (i.e. reduced torque output) should have resulted in longer fascicle lengths with the fascicles then operating further up the ascending limb compared with REF. As operating closer to the plateau results in larger force and torque capacity, the observed rFD might be underestimated. Similarly, the non-significant rFD following SHO might be explained by longer fascicle lengths compared with REF.”

Line 558-566

Referee #2:

We thank referee #2 for reviewing our manuscript and also for the constructive criticism provided. Please find our point-by-point responses to your concerns below.

"The stretch-shortening cycle effect is not associated with cortical or spinal excitability modulations" study sought to investigate the contributions of cortical and spinal excitability to force production in the stretch shortening cycle (SSC) relative to a pure shortening (SHO) contractions. The overall conclusion was the SSC is neither associated with cortical or spinal excitability. Although the paper is well-written the challenge is the complexity of using 5 different conditions to address the identified hypothesis of greater torque output from SSC compared with shortening (SHO) alone. The reader has to 'hunt' and somewhat assume the necessity for all conditions. Although Fig 3 is nicely laid out, and the electrical stimulation and protocol described the benefit of multiple conditions to address a comparative hypothesis should be strengthened, or the hypotheses better representative of the experimental design to strengthen the impact of this research.

We thank the referee for this comment based on which we decided to rephrase the hypotheses in a simpler way and to explain the necessity of all conditions in an easier way so that the readers do not have to hunt for the necessity of all conditions by themselves.

"We expected that (1) the stretch in our SSC condition would lead to tFE and rFE and facilitate the SSC effect, therefore we recorded torque following stretch and during the steady state of reference (REF_{DF}) and pure stretch (STR). We further expected (2) an increased torque during the shortening phase of SSC compared with SHO and reduced or similar rFD during the steady state following the shortening phase of SSC compared with SHO when the soleus EMG amplitudes were matched. Finally, we expected that (3) possible stretch-triggered changes in cortical and/or spinal excitability would be long-lasting and that these changes would be visible during the shortening and steady-state phases of SSC compared with SHO and reference (REF_{PF}) when EMG amplitudes were matched."

Lines 123-131

Authors from this study have previously reported (for example Hahn et al. 2012) larger motor evoked potentials (MEPs) in the presence of rFE during maximal voluntary plantar flexion contractions, which was interpreted as increased cortical, but unchanged spinal excitability (lines 95-98). This is an example of where some prior work is not aligning with findings from this study. This challenges the understanding of the physiological mechanisms. Addressing discrepancies directly and ensuring that all hypotheses are clearly addressed in the Discussion and in more detail (see small discussion lines 481-483 and Hyp 3 around long-lasting effects) would strengthen the paper.

The referee's concern was addressed in a new paragraph by going into more detail in the discussion about the differences between the studies and why a study from the 'same' laboratory might have generated results that are contradictory:

"This interpretation is in contrast with the study by Hahn et al. (2012) [...] also contrary to Sypkes et al. (2018a) [...]. A direct comparison of our results with these two studies should be made with caution given the different muscle groups and muscle lengths tested at different

levels of voluntary effort. In addition, the large variability of MEPs has already been extensively discussed by Frischholz et al., (2022). Instead, by averaging 20 MEPs (Brownstein et al., 2018) in a larger sample of 14 participants, Frischholz et al. (2022) found that corticospinal excitability remained unchanged during the steady-state hold phase in the presence of rFE after active muscle stretch, which is in line with our results. In addition, the use of a monophasic pulse shape (as in Hahn et al., 2012 and Sypkes et al., 2018a) or a biphasic pulse shape (as in Frischholz et al., 2022 and this study) to magnetically stimulate of the brain can activate different neuronal populations, which might have contributed to differences between studies (Sommer et al., 2018)."

Lines 499-516

Furthermore, based on a comment by referee 1 (Comment: Results, 2), we now state more explicitly which results are in line and which results are contradictory to our expectations/hypotheses, which should ensure that all hypotheses are clearly addressed in the Discussion.

It is difficult to understand where the Independent Variable of stimulation was tested. This might arise from confusion between using the linear mixed-effect models and ANOVAs (eg., two-way ANOVA; contraction condition x muscle). Line 361 identifies no effect of the stimulation technique for what seems to be an ANOVA. Identify and justify why linear mixed-effect models as well as ANOVAs were used, and how the stimulation conditions were tested (they are identified in the regressions). The statistics also fail to identify if the data was normally distributed, and thus the appropriacy of a parametric design cannot be assessed. The number of data points in the regressions are highly variable, explain how the data was included and/or excluded and how this influences the power of the statistics.

Thank you for this comment. We noticed that the description of the statistics was partly incomplete, which certainly contributed to the confusion. We now added the following sentence:

„We used a two-way repeated-measures restricted maximum likelihood mixed-effects model [...] to assess mean differences in resultant torque (five missing values) among the contraction conditions and stimulation techniques (contraction condition x stimulation technique)."

Lines 329-335

We first tested whether torque was affected by the stimulation techniques used. That's why we chose a two-way repeated-measures restricted maximum likelihood mixed-effects model. As we observed no significant main effect of stimulation technique ($p=0.422$) and no significant interaction between stimulation technique and contraction condition ($p=0.652$) on torque, we then averaged the torque of all trials from the same condition irrespective of stimulation technique. Subsequently, we performed a paired t-test to compare torques between SHO and SSC during shortening. Further, we performed a one-way repeated-measures ANOVA to compare torques between REF_{PF}, SHO and SSC during the steady-state following shortening.

Thus, the results section has now been corrected as follows:

“Torque was not significantly affected by stimulation technique (shortening, $F_{1.22, 13.48} = 0.76$, $p = 0.422$; steady state, $F_{1.52, 21.35} = 0.40$, $p = 0.618$), with no significant interaction between stimulation technique and contraction condition (shortening, $F_{1.35, 11.55} = 0.31$, $p = 0.652$; steady state $F_{2.71, 37.92} = 1.20$, $p = 0.313$).”

Lines 371-374

Linear mixed-effect models as well as ANOVAs were used since Prism offers the fitting of a mixed effects model to analyze repeated measures data with missing values. Thus, the mixed-effects model produces the same results as repeated measures ANOVA when there are no missing values or when participants were missing completely, and comparable results when there are missing values. Therefore, the use of repeated measures ANOVA or mixed-effects models was dependent on whether there were missing values.

Further, it has been shown that for repeated measures ANOVA, the type I error and power are not affected by violations of normality (Blanca et al., 2023). Therefore, there is no need to use a non-parametric test when normality is violated unless there are extreme outliers affecting the mean. We therefore decided against testing for a normal distribution in advance before performing a repeated measures ANOVA or a linear mixed-effects model. However, since this does not apply to the paired t-test, we had to check the normal distribution retrospectively for the comparison of the torque data for peak torque (tFE) and torque during shortening (SHO vs. SSC) and torque in the steady state after the pure stretch (REF_{DF} vs. STR). While the normal distribution was given for peak torque (tFE) and torque during shortening (SHO vs. SSC), it was violated for REF_{DF} vs. STR. As a result, we performed a non-parametric test (Wilcoxon test) for the latter and still obtained a significant result.

This has now been explained as follows:

“[...] a Wilcoxon test was used to determine whether there was a significant mean difference in plantar flexion torque during the steady state between REF_{DF} and STR, since a normal distribution of the paired differences was violated as indicated by a Shapiro-Wilk test.”

Lines 344-347

Data was included based on (1) the EMG_{SOL} matching criterion, as described in line 245 and/or (2) the cross-correlation of the response shape ($r > 0.8$) between the averaged trials and the individual trials of each contraction condition for each muscle separately (lines 285-287). If the matching criteria described in the text were not met, the respective trials were excluded, resulting in the varying number of data points.

Our sample was reduced from $n=15$ to $n=12$ for torque, from $n=15$ to $n=14$ (during shortening) and $n=13$ (during steady state) for MEPs, and from $n=15$ to $n=13$ (during shortening) and $n=12$ (during steady state) for CMEPs. If the exclusion criteria were not considered, this would possibly have led to data being included that is not comparable and would therefore distort the overall findings. We have already tried to avoid losing statistical power due to data exclusion with a larger sample tested than previously calculated ($n=18$ instead of $n=15$). However, due to the complexity and the time required for this experiment, it was not possible to collect any more data, but we assume that the use of these strict selection criteria is more likely to strengthen the validity of our study, especially with regard to the detection method used to identify valid responses.

Line 126: define SHO on first use

We thank the referee for picking this up and defined SHO.

“[...] shortening contractions without a preceding stretch (SHO) [...]”

Lines 59

Line 134: likely insufficient data to consider sex-differences. However, have two different shapes for females and males in figures and/or discuss potential, through observation, how sex MIGHT be influencing the findings.

Although we agree with the referee that it is important to consider potential sex differences, based on muscle fibre experiments we did not expect such differences (Fukutani and Herzog, 2019, Tomalka et al. 2020, 2021). Accordingly, our study was neither designed nor powered to detect potential sex-differences.

However, to doublecheck for potential sex-differences, we performed a two-way ANOVA or repeated-measures restricted maximum likelihood mixed-effects model to assess mean differences in resultant torque among the contraction conditions and between sexes (contraction condition x sex) for the SSC-effect, rFD and rFE. There was no significant interaction between contraction condition and sex for the SSC-effect ($F_{1,9} = 0.22, p = 0.649$), rFD ($F_{2,16} = 0.37, p = 0.690$), or rFE ($F_{1,9} = 0.93, p = 0.357$). For this reason, we assume that no sex differences exist in our data set; however, as mentioned above, this could also be because of the small sample size (male = 10 and female = 5 for the included data) and low power for such a comparison.

Therefore, we would like to not include any sex comparison in the manuscript. Nonetheless, all data will be made available so that readers could test for such differences themselves.

Line 210: clarify hotspot marking on skin, notably in females where this can be problematic for repeatability.

Based on the referee's comment, we clarified the hotspot marking on the skin:

“The hotspot was then marked on the skin with a stencil marker (Purple Stencil Marker™ Surgical Skin Scribe, Stencil Stuff, USA) for replication throughout the experiment. Replication was achieved by drawing the outer shape of the coil on the scalp. This created a V shape measuring approximately 5 cm, which made it possible to reposition the coil as accurately as possible on the head.”

Lines 212-216

However, experience has shown that it was no problem to mark the hotspot on females. We assume that this comment more so refers to longer hair. But as soon as the hotspot was clearly marked and the hair was parted accordingly, reproducibility was no longer a problem.

Line 231: the 't' is missing in sTimulation

Thank you for spotting this.

Line 286-288: It seems like these are the minimum number of potentials that are included in the mean data. Report the average and/or range of what was included in the Results.

As the referee correctly noted, we have only reported the minimum number of potentials that are included in the mean data. As suggested by the referee, we now also report the range of what was included in the results:

“Resulting in the inclusion of always 3 Mmax, 3-5 CMEP and 5-10 MEP responses per contraction condition per participant.”

Lines 293-294

Line 393-394: Be specific on the significant effect found

We changed the manuscript according to the referee’s suggestion:

“We found a significant SSC effect of ~12.0 % [...]”.

Line 411

Line 403: Consider rephrasing, as "We have termed this as..." makes it seem like this is the first use of these terms. Provide references.

Thank you for pointing this out. We changed the text and provided a reference.

“[...] which was termed tFE and rFE, respectively (Bakenecker et al., 2020).”

Line 419

However, we would like to mention that Bakenecker et al. (2020) is the first paper to refer to these terms in the context of tFE and rFE as such - therefore this paper is used as a reference. Other authors used different terms and abbreviations for “tFE”, thus it is difficult to determine who first coined the term.

Line 404: Incomplete sentence

Thank you for picking this up. The sentence was completed and rephrased:

“We performed a control comparison (REF_{DF} vs. STR) to assess whether our stretch conditions elicited tFE and rFE. As expected, we found significant tFE of $15 \pm 25\%$ at the end of stretch and significant rFE of $7 \pm 3\%$ following stretch relative to REF_{DF} .”

Line 419 - 422

Figures: consider using shapes and/or colors more systematically and in so doing identify the sex of the participants to increase robust reporting and contribute to understanding sex-differences in these contractions.

We have already used colors systematically to distinguish each participant, and we have used shapes systematically to distinguish between the comparison of the different contractions conditions for the correlations performed (Figure 4-6). We think that identifying the sex of the participants adds unnecessary complexity as our study was not powered to look at sex differences.

We hope that the referee agrees with our argumentation.

References

- Blanca, M. J., Arnau, J., García-Castro, F. J., Alarcón, R., & Bono, R. (2023a). Non-normal data in repeated measures: Impact on Type I error and power. *Psicothema*, **35**(1), 21–29.
- Fukutani, A., Joumaa, V., & Herzog, W. (2017). Influence of residual force enhancement and elongation of attached cross-bridges on stretch-shortening cycle in skinned muscle fibers. *Physiological reports*, **5**(22), e13477.
- Fukutani, A., & Herzog, W. (2019). Influence of stretch magnitude on the stretch–shortening cycle in skinned muscle fibres. *Journal of Experimental Biology*, **222**(13), jeb206557.
- Fouré, A., Cornu, C., McNair, P. J., & Nordez, A. (2012). Gender differences in both active and passive parts of the plantar flexors series elastic component stiffness and geometrical parameters of the muscle–tendon complex. *Journal of Orthopaedic Research*, **30**(5), 707–712.
- Goecking, T., Holzer, D., Hahn, D., Siebert, T., & Seiberl, W. (2024). Unlocking the benefit of active stretch: The Eccentric muscle action not the preload maximizes muscle-tendon unit stretch-shortening cycle performance. *Journal of Applied Physiology*.
- Holzer, D., Hahn, D., Schwirtz, A., Siebert, T., & Seiberl, W. (2024). Decoupling of muscle tendon unit and fascicle velocity contributes to the in vivo stretch-shortening cycle effect in the male human triceps surae muscle. *Physiological Reports*, **12**(23), e70131.
- Raiteri, B. J., Lauret, L., & Hahn, D. (2024). Residual force depression is not related to positive muscle fascicle work during submaximal voluntary dorsiflexion contractions in humans. *The Journal of Physiology*, **602**(6), 1085–1103.
- Tomalka, A., Weidner, S., Hahn, D., Seiberl, W., & Siebert, T. (2020). Cross-bridges and sarcomeric non-cross-bridge structures contribute to increased work in stretch-shortening cycles. *Frontiers in physiology*, **11**, 921.
- Tomalka, A., Weidner, S., Hahn, D., Seiberl, W., & Siebert, T. (2021). Power amplification increases with contraction velocity during stretch-shortening cycles of skinned muscle fibers. *Frontiers in physiology*, **12**, 644981.

Dear Professor Hahn,

Re: JP-RP-2025-287508R1 "The stretch-shortening cycle effect is not associated with cortical or spinal excitability modulations" by Lea-Fedia Rissmann, Daniel Hahn, Brent James Raiteri, Tobias Siebert, and Wolfgang Seiberl

We are pleased to tell you that your paper has been accepted for publication in The Journal of Physiology.

Yours sincerely,

Richard Carson
Senior Editor
The Journal of Physiology

If you would like to receive our 'Research Roundup', a monthly newsletter highlighting the cutting-edge research published in The Physiological Society's family of journals (The Journal of Physiology, Experimental Physiology, Physiological Reports, The Journal of Nutritional Physiology and The Journal of Precision Medicine: Health and Disease), please click this link, fill in your name and email address and select 'Research Roundup':
<https://www.physoc.org/journals-and-media/membernews>

- **TRANSPARENT PEER REVIEW POLICY:** To improve the transparency of its peer review process, The Journal of Physiology publishes online as supporting information the peer review history of all articles accepted for publication. Readers will have access to decision letters, including Editors' comments and referee reports, for each version of the manuscript as well as any author responses to peer review comments. Referees can decide whether or not they wish to be named on the peer review history document.
- You can help your research get the attention it deserves! Check out Wiley's free Promotion Guide for best-practice recommendations for promoting your work at: www.wileyauthors.com/eeo/guide. You can learn more about Wiley Editing Services which offers professional video, design, and writing services to create shareable video abstracts, infographics, conference posters, lay summaries, and research news stories for your research at: www.wileyauthors.com/eeo/promotion.
- **IMPORTANT NOTICE ABOUT OPEN ACCESS:** To assist authors whose funding agencies mandate public access to published research findings sooner than 12 months after publication, The Journal of Physiology allows authors to pay an Open Access (OA) fee to have their papers made freely available immediately on publication.

EDITOR COMMENTS

Reviewing Editor:

Thank you for considering and responding to the comments made by the reviewers. The manuscript reads very well and presents some novel and interesting findings. While it was highlighted by one reviewer that you may want to consider including more detail around the sex differences in the figures given the lack of power to address this in detail and the

availability of data should anyone wish to consider this further in this instance you do not need to address this comment.

REFEREE COMMENTS

Referee #1:

The authors have addressed my concerns. I have no further comments.

Referee #2:

Thank you for the thorough and detailed responses that address prior comments.

I agree with the authors that the study is not powered to consider sex. However, I disagree on the value, and the ease in which sex can be identified within a paper, or supplementary material. Ultimately this decision belongs with the journal, as it must align with their practises. However, to advance physiology the sex of the participant needs to be considered. This does not suggest that sex differences will exist in all aspects. However, the lack of sex difference also contributes to knowledge. Without reporting the sex, and data for sex future work, such as meta analysis is not as easily achieved. Please consider the broader implications of reporting sex, event when the particular research undertaken is not powered, or considering a sex-based approach <https://pmc.ncbi.nlm.nih.gov/articles/PMC10694598/> that align with recent advances in scientific approaches.